# A brain-specific *pgc1α* fusion transcript affects gene expression and behavioural outcomes in mice

Oswaldo A Lozoya[1], Fuhua Xu[1], Dagoberto Grenet[1], Tianyuan Wang[2], Korey D Stevanovic[3] , Jesse D Cushman[3], Thomas B Hagler[4], Artiom Gruzdev[4] , Patricia Jensen[5], Bairon Hernandez[6], Gonzalo Riadi[6], Sheryl S Moy[7], Janine H Santos[1] , Richard P Woychik[1]

**PGC1α is a transcriptional coactivator in peripheral tissues, but its function in the brain remains poorly understood. Various brain-specific *Pgc1α* isoforms have been reported in mice and humans, including two fusion transcripts (FTs) with non-coding repetitive sequences, but their function is unknown. The FTs initiate at a simple sequence repeat locus ~570 Kb upstream from the reference promoter; one also includes a portion of a short interspersed nuclear element (SINE). Using publicly available genomics data, here we show that the SINE FT is the predominant form of *Pgc1α* in neurons. Furthermore, mutation of the SINE in mice leads to altered behavioural phenotypes and significant up-regulation of genes in the female, but not male, cerebellum. Surprisingly, these genes are largely involved in neurotransmission, having poor association with the classical mitochondrial or antioxidant programs. These data expand our knowledge on the role of *Pgc1α* in neuronal physiology and suggest that different isoforms may have distinct functions. They also highlight the need for further studies before modulating levels of *Pgc1α* in the brain for therapeutic purposes.**

## Introduction

There is increasing interest in the role of the peroxisome proliferator-activated receptor γ coactivator 1 α (PGC1α) in the brain, given mounting evidence that its levels are modulated in various neurodegenerative disorders including Huntington's (HD), Parkinson's (PD), and Alzheimer's disease (AD) as well as amyotrophic lateral sclerosis (Katsouri et al, 2012; Dumont et al, 2014). In skeletal muscle, liver, heart, and brown adipose tissue, PGC1α co-activates a series of genes prominently associated with mitochondria biogenesis, lipid metabolism, antioxidant defences, and thermogenesis (Lin et al, 2004).

Given the importance of mitochondrial biogenesis and renewal to proper neuronal function, it was rather surprising to find that the conditional deletion of *Pgc1α* in the central nervous system (CNS) led to only modest changes in these processes in mice. Instead, only a few genes associated with brain-specific functions were found in the animals (Cui et al, 2006; Lucas et al, 2010, 2012, 2014b; McMeekin et al, 2018), suggesting that PGC1α in the brain has different roles, or downstream targets, than in peripheral tissues. Adding to this conundrum are data reporting the existence of multiple isoforms of *Pgc1α* in the CNS (Soyal et al, 2012), raising questions of whether the canonical/reference transcript is the most relevant to brain physiology.

In our previous work, we identified two mouse brain fusion isoforms of *Pgc1α* that initiated transcription from a promoter located 566.7 Kb upstream from exon 2 (Wang et al, 2016). At this position, we found a simple sequence repeat (SSR) that spliced directly to the second common coding exon of *Pgc1α* (SSR-exon 2); we also found a second fusion transcript (FT) in which the SSR spliced to a portion of a short interspersed nuclear element (SINE) about 200 Kb downstream of it, which then spliced to exon 2 (SSR-SINE-exon 2) (Fig 1A). We validated the expression of these isoforms in the brain by RT-PCR and found, using publicly available RNA-seq, that the SINE-containing FT was more abundant than the reference isoform in the ventral tegmental area, amygdala, hippocampus, and prefrontal cortex. Also, we found that these transcripts were brain-specific, and that the SSR and the SINE were conserved in rodents, humans, non-human primates, dogs, chickens, and sticklebacks (Wang et al, 2016). Based on genomic coordinates, the SSR and SINE exons seem to correspond to the previously described human B1 and B4 exons of a human CNS-specific isoform of *Pgc1α* that was associated with HD (Soyal et al, 2012). However, it remains unknown whether the B1-B4 isoform is functional.

The B1 promoter of the CNS-specific *Pgc1α* was reportedly located ~587 kb upstream of exon 2 in humans. Expression levels of

[1]Genomic Integrity and Structural Biology Laboratory, National Institutes of Health, Durham, NC, USA   [2]Integrative Bioinformatics Branch, National Institutes of Health, Durham, NC, USA   [3]Neurobehavioral Core Laboratory, National Institutes of Health, Durham, NC, USA   [4]Knockout Mouse Core Facility, National Institutes of Health, Durham, NC, USA   [5]Neurobiology Laboratory, National Institute of Environmental Health Sciences, National Institutes of Health, Durham, NC, USA   [6]Centro de Bioinformática y Simulación Molecular, Facultad de Ingeniería, Universidad de Talca, Talca, Chile   [7]Department of Psychiatry, Carolina Institute for Developmental Disabilities, University of North Carolina at Chapel Hill, Chapel Hill, NC, USA

Correspondence: janine.santos@nih.gov; rick.woychik@nih.gov

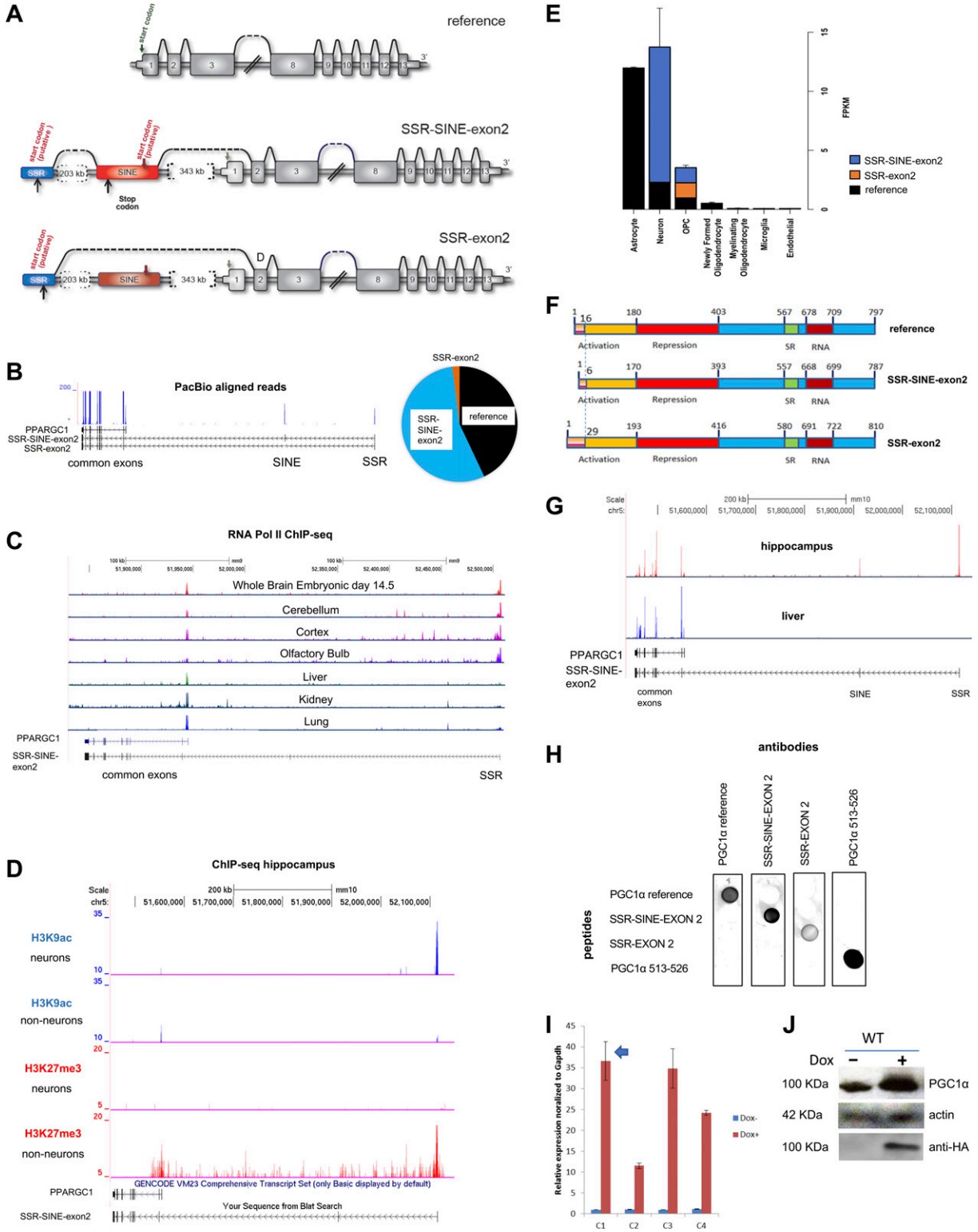

**Figure 1. Novel isoforms of short interspersed nuclear element (SINE)-containing *Pgc1α* are expressed in the mouse brain.**

**(A)** Schematic representation of gene structure of the reference, simple sequence repeat (SSR)-SINE-exon 2 and SSR-exon 2 isoforms of *Pgc1α*. Brackets depict distance between the exons, black line on top indicates splicing, grey arrow over exon 1 indicates reference start codon. SSR exon = 448 bp, putative ATG at 362–364 (indicated by black arrow below it); TGA is located at position 19–21 of the SINE (black arrow below it) with the ATG (red arrow above it) at position 48–50. The SINE exon is 65 bases long. **(B)** UCSC browser track depicting the genomic location of SSR, SINE, and exons of the reference *Pgc1α* gene (in black); gene goes from left to right. Common exons reflect those present in all isoforms, excluding exon 1 that is only present in the reference gene. In blue are PacBio peaks from reads aligning to each respective exon of the gene. Pie chart on the right depicts the proportion of full-length reads that covered the entirety of each isoform as shown in Table S1; only data obtained using reads >2 Kb are

the isoforms originating from B1 were similar or higher than that of the annotated *Pgc1α* gene, similar to our FTs, and were confined to specific cell types. For example, whereas astrocytes expressed the transcript originating from the reference promoter, neurons and oligodendrocytes transcribed primarily the isoform that initiated from B1 and that contained exon B4 (Soyal et al, 2012). The B1-B4–containing isoform was found to be up-regulated in the striatum, cortex and cerebellum of mice treated with 1-methyl-4-phenyl-1,2,3,6-tetrahydropyridine (MPTP), a drug commonly used to model PD (Torok et al, 2017). Conversely, the B1-B4 isoform was identified as being down-regulated in amyotrophic lateral sclerosis mouse models (Bayer et al, 2017). More recently, the B1 exon, but not the reference *Pgc1α* promoter, was shown to be activated in vitro by HIF1α (Soyal et al, 2020), leading to the hypothesis that the brain isoforms may have distinct transcriptional regulation. Last, haplotypes encompassing the human region of the B1 promoter have been associated with the age of onset of HD (Soyal et al, 2012) and with protection against PD (Soyal et al, 2019), collectively suggesting that sequence variations in the brain isoforms of *Pgc1α* may contribute to disease.

In this work, we first functionalize the SINE-containing FT isoform, which based on genomic coordinates is the homologue of the B1-B4–containing transcript. We then tested the hypothesis that the SINE FT has its own set of targets in the brain. By generating a mouse carrying a mutation on the SINE while leaving the reference *Pgc1α* isoform intact, here we show that loss of the SINE FT leads to impaired motor coordination. Most importantly, its absence is paralleled by the up-regulation of hundreds of genes in the female but not male cerebellum, which do not involve the classical mitochondrial or antioxidant programs but rather neurotransmission. These data place the SSR-SINE-exon 2 isoform of *Pgc1α* as relevant to brain physiology in ways that are distinct from the role of the reference gene.

## Results

### Novel brain-specific *Pgc1α* isoforms are produced from a promoter in the SSR

The ~600 Kb pre-mRNA of our previously identified brain FT isoforms of *Pgc1α* (Fig 1A) is predicted to require between 3.3 and 10 h to be transcribed, assuming an average transcription rate of 1–3 Kb/min (Wada et al, 2009). Although it is not unusual for brain transcripts to be exceedingly large (Zylka et al, 2015), the first step in our

studies was to confirm that the full-length mRNAs existed in vivo. We used PacBio Technology, which generates sequencing reads of up to 60 Kb in length (Rhoads & Au, 2015), to determine the types of *Pgc1α* mRNA present in the whole mouse brain. Read lengths obtained under our experimental conditions ranged from 500 bp to >5.5 Kb (Table S1). We initially analysed reads >2 Kb because they would encompass the entire mRNA of the predicted novel isoforms. We found evidence for transcription of the reference, SSR-exon 2 and the SSR-SINE-exon 2 isoforms, with the latter being the most abundant (Fig 1B and Table S1). No reads containing sequences upstream from the SSR were identified. When shorter reads were analysed (see the Materials and Methods section), we found additional evidence for the presence of the SINE isoform as well as other non-canonical exon–exon pairs (for details see Table S1), suggesting that there may be additional as yet uncharacterized isoforms in the brain that could be analysed.

Having confirmed that full-length transcripts occurred in vivo, we next determined if the SSR locus contained the promoter. To this end, we mined publicly available chromatin immunoprecipitation sequencing (ChIP-seq) data for RNA polymerase II (RNA Pol II) and for the histone H3K9ac mark (ENCODE LICR-TFBS; Centeno et al, 2016), both of which are characteristically enriched at promoter regions. We found RNA Pol II peaks at the SSR locus in the brain (whole brain, cortex and cerebellum) but not in the liver, kidney or lung where peaks mapped to the reference *Pgc1α* promoter (Fig 1C). The olfactory bulb also showed an RNA Pol II peak over the SSR (Fig 1C). Likewise, ChIP-seq data from the hippocampus showed that H3K9ac peaks were prominent over the SSR locus in neurons but not in non-neuronal cells, which were enriched for the repressive H3K27me3 mark (Fig 1D). These results show that the SSR region contains epigenetic marks normally associated with regulation of transcriptional initiation. It is noteworthy that the SSR genomic coordinates coincide with a CpG island, which is frequently found associated with promoters in mammals, further supporting the notion that it contains the promoter of the novel *Pgc1α* brain isoforms. The H3K9ac data also suggest that the promoter at the SSR locus is primarily responsible for transcription of *Pgc1α* in neurons.

The above data prompted us to define whether expression of the different *Pgc1α* transcripts is cell type-specific in the brain. To address this, we used RNA-seq derived from distinct brain cell types (Zhang et al, 2014) and compared the abundance of reads spanning the junctions between the SSR-exon 2, SINE-exon 2 and exon 1-exon 2 to estimate the expression levels of the FTs and the reference *Pgc1α*. Although several brain cell types were present in the dataset

represented. **(C)** ChIP-seq peaks of RNA polymerase II (pol2) over the coordinates of the SSR or exon 1 of *Pgc1α* in different tissues. The genomic coordinates of the reference isoform are shown in blue on the bottom left corner. **(D)** Same as C but using ChIP-seq data for the promoter H3K9ac mark (in blue) or the repressive H3K27me3 mark (red) in the hippocampus. **(E)** Number of RNA-seq counts covering the junctions of SSR-exon 2, SINE-exon 2 or exon 1-exon 2 were used to establish the degree of expression of each of the three major isoforms of *Pgc1α* in brain-specific cell types. Data are depicted as counts per FPKM (fragments per Kb per million). **(F)** Schematic representation of the putative protein structures of the two novel brain isoforms of *Pgc1α*; the protein derived from the reference gene is also depicted. Numbers above reflect amino acid positions; known domains are shown below. SR, serine–arginine rich. **(G)**. Ribo-seq data from hippocampus (red) and liver (blue) were used to define the presence of the SSR, SINE and exons of reference *Pgc1α* within actively translating ribosomes. Genomic structure of the reference and SSR-SINE-exon 2 isoforms are shown in black below. **(H)** Antibodies were generated against epitopes for the SSR, SINE, exon 1-2 junction and the C-terminus (amino acids 513–526) of PGC1α. Peptides used to generate the four different antibodies were spotted on each membrane strip and incubated with the serum separately. **(I)** NIH3T3 cells were transfected with a doxycycline-inducible HA-tagged vector expressing the reference isoform of *Pgc1α*. Quantitative RT-PCR was used to estimate the expression level of *Pgc1α* among different clones; data were normalized to GAPDH. **(J)** Clone 1 (from B) was selected to test antibody specificity; only the antibody against the C-terminus detected a band of the same size as that detected with an anti-HA antibody.

(Fig 1E), only those with significant *Pgc1α* expression were considered for the analysis. We found that astrocytes expressed primarily the reference isoform because all reads from *Pgc1α* spanned the junctions between exons 1 and 2. Conversely, most reads covered the SINE-exon 2 junction in neurons, whereas in oligodendrocyte progenitor cells, junction reads corresponding to all three isoforms were identified in similar proportions (Fig 1E). Thus, distinct cell types express different isoforms of *Pgc1α* in the mouse brain. Most importantly, the SSR-SINE-exon 2 transcript seems the primary isoform expressed in neurons.

### The SSR-SINE-exon 2 isoform of Pgc1α is translated into protein

The N-terminus of PGC1α is thought to dictate its transcriptional targets (Soyal et al, 2012; Martinez-Redondo et al, 2015), although whether the C-terminus can also influence gene expression has not been fully ruled out (Sadana & Park, 2007). Both FTs skip exon 1 where the ATG used for translation initiation of the reference *Pgc1α* transcript is present. Thus, these isoforms would need to use alternative ATGs if translated. In turn, they would give rise to proteins with different N-termini or reading frames. Analysis of the 5′ sequences of the new *Pgc1α* mouse FTs revealed an alternative ATG within the SSR (Fig 1A), which connects with the downstream exons through exon 2; this would give rise to a protein 810 amino acids long and with 29 novel residues at its N-terminus (Fig 1F). A stop codon within the SINE makes it likely that the second ATG found within this exon initiates translation of the FT, running through the downstream ORF (Fig 1A). The resulting protein would have six SINE-encoded amino acids that replace the 16 amino acids from exon 1 at the N terminus of the reference protein (Fig 1F).

Next, we obtained evidence for translation of the FTs by using publicly available ribosomal profiling data. Ribo-seq or ribosomal foot printing relies on deep sequencing of mRNA molecules after immunoprecipitation of ribosomes, giving a snapshot of transcripts actively translated within a cell (Ingolia, 2014). Thus, if the FT isoforms are translated into protein, the SSR and SINE should be captured in the Ribo-seq dataset as part of the *Pgc1α* transcript. We mined data derived from the hippocampus (Cho et al, 2015) and the liver (Howard et al, 2013), with the latter serving as negative control. Consistent with the FTs being translated, large peaks were detected over the coordinates of the SSR, SINE and other exons from *Pgc1α* starting from exon 2 in the hippocampus (Fig 1G). Conversely, peaks covered only exon 1 of the reference isoform in the liver, with no peaks over the coordinates of either the SSR or SINE (Fig 1G, compare red and blue lines), overall demonstrating that the FTs are translated in the brain but not in the liver. To further confirm these findings, we developed antibodies against epitopes unique to the predicted novel proteins: for the SSR-exon 2, we used amino acids that crossed the SSR-exon 2 junction. For SSR-SINE-exon 2, we used the unique six amino acids coded by the SINE; we also use amino acids 513–526 at the C-terminus given this part of the gene is common to all PGC1α isoforms. Antibodies were highly specific to the peptides they were developed against (Fig 1H), but in tissue lysates those raised against the FTs were unable to detect one only protein. Antibodies for the C-terminus recognized a protein of the correct molecular weight of an engineered HA-tagged PGC1α recombinant protein that we expressed in NIH3T3 cells (Fig 1I and J).

This antibody was later used to define the absence of the SINE-derived protein in the brain by assuming that a decrease in the total levels of PGC1α would reflect loss of the SINE isoform (see below).

### Mutation of the SINE in mice preserves normal brain anatomy but impairs behaviour and motor performance

Despite work characterizing the expression or regulation of the B1-B4 isoform of *Pgc1α* in brain (Soyal et al, 2012, 2020), to date there is no published evidence that it is functional in vivo. PGC1α KO mice, either full body or CNS-specific, have been created (Lin et al, 2004, Leone et al, 2005, Lucas et al, 2012, 2014a, 2014b) but deletion of the common exon 3 precludes interpretation about isoform-specific functions. Thus, to make a mouse model that could adequately establish the functional significance of the SSR-SINE-exon 2, we used CRISPR/Cas9 to target the SINE. We established a line of mutant mice with a 4-bp intragenic deletion immediately downstream of the putative ATG within the SINE (Fig S1), which is predicted to abort translation of the SSR-SINE-exon 2 transcript. We confirmed that the transcript was still present in the brain of mutant animals and that no compensatory changes occurred in the expression of the reference isoform (Fig S2A and B). Using the antibody generated against C-terminus of PGC1α (Fig 1H–J), we found a band in the brain of the wild-type (WT) animals with the same molecular weight as the HA-tagged PGC1α (Fig 2A). In mutant littermates, only a faint band was present in the brain, which is expected given the protein derived from the reference PGC1α isoform is not affected in our animals (Fig 2A). This antibody detected a protein in the liver of both the WT and homozygous mutant animals, consistent with the SINE-containing RNA being a brain-specific isoform (Fig 2A). Thus, while the transcript containing the 4-bp deletion of the SINE isoform is present, we interpret the significant reduction in total PGC1α protein in the brain of mutant animals to mean that the protein derived from this isoform is not translated as predicted, making them functional KOs. We thus refer to these animals as SINE KO mutants.

Maintenance of this line revealed that homozygous mutant pups were born at the expected Mendelian ratio although an increase in the number of heterozygotes was noted in females (Fig S2C). No postnatal lethality was observed, in contrast to that reported with the exon 3 deletion mutants (Lin et al, 2004). It has been proposed that such lethality was associated with a reduction in fat differentiation and hypothermia resulting from the loss of PGC1α in peripheral tissues, which is maintained in our animals (Fig 2A). Also, unlike for the exon 3 deletion allele, SINE KO homozygotes showed no significant changes in body weight of males or females until later in life when males were ~10% leaner (Fig 2B). As shown in Fig S2D, cresyl violet staining of sagittal brain sections did not reveal any gross anatomical abnormalities, which are hallmark features of animals carrying alleles associated with the exon 3 deletion (Lin et al, 2004). Likewise, we did not detect the reported spongiform lesions in the striatum or pathology in the cerebellum (Fig S2E), nor did we observe reduced locomotion, muscle weakness or ataxia-associated signs as described previously for the exon 3 allele (Lin et al, 2004). Thus, elimination of the PGC1α protein expressed from the SSR-SINE-exon 2 isoform does not contribute to the postnatal lethality, neuropathology, muscle weakness, and ataxia previously

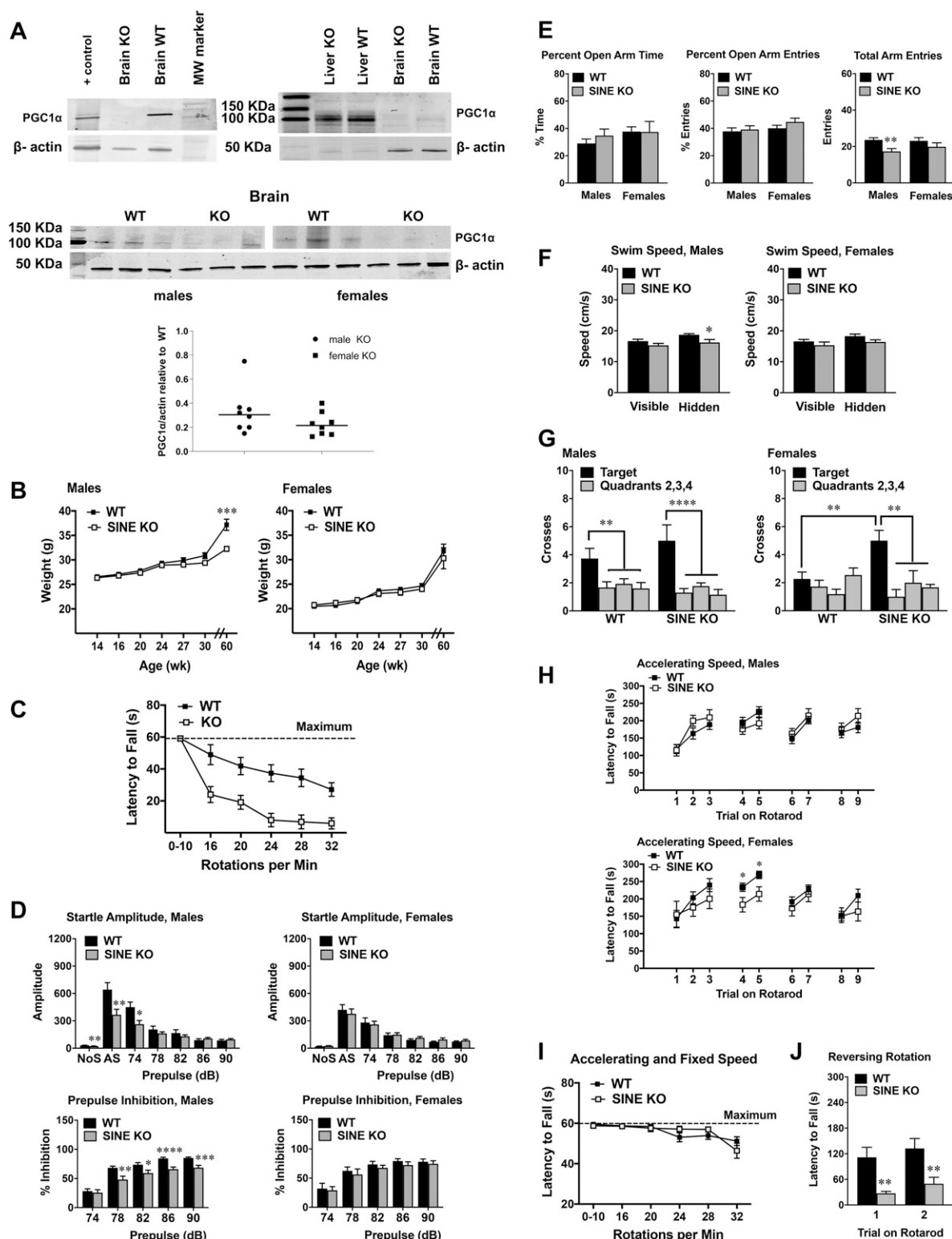

**Figure 2. Mutations on the short interspersed nuclear element (SINE) lead to sexual dimorphic behavioural deficits in the absence of gross brain lesions.**
**(A)** Western blots in brain lysates from WT or KO animals (left panel); lysates from HA-tagged PGC1α-overexpressing NIH3T3 were used as positive control. Right panel shows data from liver and brain lysates. Lower panel show additional samples from animals that have submitted to the behavioural tests; male and female samples were run in separate gels for PGC1α, whereas all 12 samples were run together for actin. Graph depicts the mean PGC1α relative to actin in KO animals relative to the WT counterparts (n = 8/sex/group). **(B)** Weight from WT and SINE KO over time; final weight measure was approximately at 60 wk in age. Male mice (genotype × age interaction, F[6,156] = 10.97, P < 0.0001); ***P = 0.0009. **(C)** Rotarod test in which mice underwent 2 trials each at rotating speeds of 0–10 accelerating and 16, 20, 24, 28, and 32 fixed RPM. Each trial was a maximum of 60 s, with at least 5 min between each trial (KO n = 18, 12 males and 6 females, WT n = 15, 8 males and 6 females). Data are means

reported when all brain isoforms of PGC1α are ablated through the deletion of exon 3. Given the SINE FT is the primary neuronal transcript, those phenotypes seem associated with the loss of PGC1α in non-neuronal cell types. This conclusion is in line with a recent study that proposed that the neurological phenotypes are oligodendroglial in origin (Szalardy et al, 2016).

Behavioural changes including hyperactivity and severe impaired motor coordination in the rotarod test were associated with the brain lesions found in the exon 3 deletion alleles (Lucas et al, 2012). To determine the contribution of the SINE FT isoform to this phenotype, we subjected SINE KO homozygotes and their WT littermates to the same rotarod assay applied by Lucas et al (2012). The test involved two trials, each of increasing fixed rotational speeds from 16 to 32 RPM. SINE mutants performed poorly in this test (Fig 2C), essentially phenocopying the defects previously described when all brain isoforms were deleted (Lucas et al, 2012). Thus, the SINE isoform is critical for motor performance in this test. We then subjected an independent cohort of animals to a battery of behavioural tests. Table 1 summarizes the timeline and overall test results obtained, which were performed sequentially. Although no statistical differences were found between WT and SINE KO homozygotes for most protocols (Fig S3A–F), a few differences were identified between the animals with sexual dimorphic trends apparent in mutant animals in some tests. For example, male but not female SINE KOs had significant decreases in the magnitude of the startle response and impaired prepulse inhibition (Fig 2D). Prepulse inhibition is disrupted in several neuropsychiatric disorders, including schizophrenia, which not only has a male preponderance but also has been associated with reduced cortical expression of PGC1α target genes (McMeekin et al, 2016). Male SINE KO homozygotes also showed decreased arm entries in the elevated plus maze (Fig 2E) and slightly slower initial swim speeds in the water maze (Fig 2F), which may be indicative of increased anxiety and impaired swimming-related motor coordination, respectively. Why the SINE KO animals can rapidly overcome the initial swim speed deficits is unclear. Interestingly, conflicting findings have been reported on anxiety-like behaviours for the full body exon 3 deletion alleles that retain an N-terminal truncated transcript (Leone et al, 2005; Szalardy et al, 2018). Finally, we found that female, but not male SINE KO homozygous mutants, had significantly higher target quadrant preferences relative to control animals in the Morris water maze (Fig 2G), suggesting improved spatial learning and memory (Vorhees & Williams, 2014).

At 16–19 wk in age, WT and SINE KO homozygous littermates were subjected to a standard accelerating rotarod protocol that differed from the test that we used initially (Fig 2C). This protocol consisted of trials in which speed progressively increased from three to a maximum of 30 RPM across 5 min. The first test consisted of three trials with 45 s in between each (Fig 2H, 1–3), which were followed by two additional trials that occurred 48 h later (Fig 2H, 4–5). In this protocol, females but not males showed significant impairments at trials 4–5 (Fig 2H). At weeks 31–36 (trials 6–7) and 50–61 (trials 8–9), these animals were retested but no differences between WT and mutant littermates were observed (trials 6–7 and 8–9, Fig 2H). These results were surprising as both males and females showed deficits in the more difficult rotarod test applied previously (Fig 2C). However, it is possible that females are more sensitive to the effects associated with the loss of the SSR-SINE-exon 2 isoform. The absence of motor deficit in the later trails (6–7 and 8–9) suggests that motor learning may be overcome in females. Notably, these same animals were submitted to the water maze test, which informs on learning and memory capabilities, a few weeks before this rotarod (Table 1).

To get more insights into the rotarod phenotype and the extent to which it involved motor learning, we took two different approaches. First, we subjected the same males and females to the more challenging rotarod test as in Fig 2C. We reasoned that if learning was involved, by having gone through a rotarod experience, the SINE mutants would perform as well as WT littermates despite the challenging protocol. Consistent with learning, no impairments were identified in mutant animals, irrespective of sex (Fig 2I). We then subjected them to a novel and more difficult protocol involving rapid reversals in directional rotation of the rotarod, which revealed deficits in both male and female mutants (Fig 2J). Given that animals were about the same age in these two tests (Table 1), the deficit in the rapid-reversal task is likely driven by difficulty and novelty rather than an age-related decline. Taken together, these data indicate that deficits in motor coordination and motor learning can be unmasked in the SINE mutants when animals are exposed to a novel and/or difficult protocol.

## SINE-mutant female mice exhibit increased neurotransmitter-associated gene expression in the cerebellum

The phenotypes observed in the rotarod tests suggest a cerebellar but not a striatal-dependent deficit because the latter is more important for motor learning (Dang et al, 2006). To gain insights into

---

(+SEM) of 2 trials per rpm. Trials 0–10 had accelerating speed; remaining trials had fixed speeds. Genotype × speed interaction: F(1,19) = 5.218, $P$ = 0.034; 16 RPM, $P$ = 0.06; 20 RPM, $P$ = 0.016; 24 RPM, $P$ < 0.001, 28 RPM, $P$ = 0.001, 32 RPM, $P$ = 0.004). **(D, E, F, G, H, I, J)** Data shown are means (±SEM) from 19 SINE KO mice (13 males and 6 females) and 26 WT controls (15 males and 11 females). **(D)** Magnitude of startle responses and prepulse inhibition in *Pgc1α* SINE isoform-specific KO mice. Trials included no stimulus (NoS) and acoustic startle stimulus (AS, 120 dB) alone. Males, amplitude, genotype × decibel interaction (F[6,156] = 7.9, $P$ < 0.0001); NoS, **$P$ = 0.006; AS, **$P$ = 0.0096; 74 dB, *$P$ = 0.0133. Males, prepulse inhibition, main effect of genotype (F[1,26] = 11.25, $P$ = 0.0025; genotype × decibel interaction, F[4,104] = 3.85, $P$ = 0.0058); 78 dB, **$P$ = 0.0028; 82 dB, *$P$ = 0.0229; 86 dB, ****$P$ < 0.0001; 90 dB, ***$P$ = 0.0002. **(E)** Elevated plus maze; total arm entries, main effect of genotype (F[1,41] = 7.89, $P$ = 0.0076); males, *$P$ = 0.0036. **(F)** Morris water maze, swim speeds on the first days of visible and hidden platform tests. Males: main effect of genotype (F[1,26] = 7.88, $P$ = 0.0094), Hidden, *$P$ = 0.0159. **(G)** Quadrant preference in the water maze during a 1-min probe trial after hidden platform training. Measures were taken of swim path crosses over the platform location (Target; Quadrant 1), and the corresponding locations in Quadrants 2, 3, and 4. Males: effect of quadrant (F[3,78] = 14.77, $P$ < 0.0001); WT, **$P$ = 0.0093; SINE KO, ****$P$ < 0.0001. Females: genotype × quadrant interaction (F[3,45] = 3.99, $P$ = 0.0133); WT versus SINE KO, **$P$ = 0.0059; SINE KO, **$P$ = 0.0045. **(H)** Motor coordination on an accelerating rotarod test. Trials 4 and 5 were given 48 h after the first three trials, when mice were 16–19 wk in age. Mice were 31–36 wk in age for trials 6 and 7 and were retested at 50–61 wk. Trials 4 and 5, females, main effect of genotype (F[1,15] = 7.41, $P$ = 0.0158); Trial 4, *$P$ = 0.04; Trial 5, *$P$ = 0.0196. **(I)** Animals were subjected to the same protocol as in (E). **(J)** At 56–66 wk, animals performed a rapid-reversal rotarod. Data from males and females are pooled. Main effect of genotype (F[1,43] = 9.49, $P$ = 0.0036); Trial 1, **$P$ = 0.0043; Trial 2, **$P$ = 0.0089.

**Table 1.   Behavioural testing regimen.**

| Age (wk) | Procedure | Outcome |
|---|---|---|
| 14–17 | Elevated plus maze test for anxiety-like behaviour | **Males—Reduced overall entries** |
| 15–18 | Locomotor activity and exploration in a 1-h open field test | No differences |
|  | Wire-hang test for grip-strength | No differences |
| 16–19 | Accelerating rotarod test | Female—impairments in trials 4 and 5 |
|  | Trials 1–3 (first test); trials 4 and 5 (second test, 48 h later) |  |
| 17–20 | Social approach in a three-chamber choice task | No differences |
| 18–21 | Marble-bury assay for anxiety and perseverative responses | No differences |
|  | Prepulse inhibition of acoustic startle responses | **Males—Reduced startle and PPI** |
| 19–24 | Buried food test for olfactory ability | No differences |
| 20–25 | Morris water maze; visible platform test | **Males—Reduced swim speed** |
| 21–27 | Morris water maze; acquisition of spatial learning | **Females—improved probe trial** |
| 23–28 | Second acoustic startle test | No differences |
|  | Hot plate test for thermal sensitivity | No differences |
| 26–31 | Conditioned fear test for contextual and cue learning | No differences |
| 28–33 | Second fear test for memory retention | No differences |
| 31–36 | Accelerating rotarod, trials 6 and 7 | No differences |
| 32–37 | Accelerating and fixed speed rotarod (1 min trials) | No differences |
| 50–61 | Accelerating rotarod, trials 8 and 9 | No differences |
| 56–66 | Rapid-reversal rotarod test (2 trials) | **Males and females—impaired** |
| 58–67 | Fixed speed rotarod (2 min trials) | No differences |

potential molecular changes in the cerebellum, we next profiled gene expression in dissected cerebella (Cb) of age-matched WT and SINE KO homozygous littermates with microarrays; the rest of the brain (Br, whole brain minus cerebellum) was included as a control. We analysed differential gene expression based on RMA-normalized probe intensities by the Leverage Signal-to-Noise Ratio (LSTNR) method (Lozoya et al, 2018) using a $2^3$ full-factorial design, n = 2 per group. Principal component analysis of 12,527 multivariate significant probe sets showed that the largest statistical differences between group means were observed in the cerebellum, most notably in females (Fig 3A). Less than 50 probes were differentially enriched in the rest of the brain for each sex (Fig 3B lower panels), whereas 2,016 probes in females and 354 probes in males were different between cerebellum of WT and SINE KO littermates (Fig 3B upper panels). The lack of broader changes in the Br likely reflects the analyses of several brain regions at once, which might be diluting region-specific effects, rather than a cerebellum-unique phenotype. Future experiments dissecting and profiling the transcriptome of individual brain regions can address this issue. Unexpectedly, most probes showing significant expression differences in female cerebella were up-regulated in the mutants (Fig 3B, upper left panel). In males, the probes that were different between WT and KO littermates were mainly down-regulated (Fig 3B, upper right panel), consistent with what is known upon loss of the reference PGC1α in peripheral tissues. RT-PCR was used to confirm the differential expression of some general targets (Fig S4A).

Microarray data did not show differences in *Pgc1α* probe intensity between the cerebellum versus rest of brain in either males or females, although a minor decrease (22%) was observed in the total levels of *Pgc1α* in the male versus the female cerebellum (Fig S4B). The lack of differential expression levels of *Pgc1α* as a function of brain region was further confirmed by RT-PCR in an independent cohort of males, in which we probed the levels of the reference gene and the SSR-SINE-exon 2 isoform (Fig S4C). Analysis of publicly available RNA-seq data from male and female brain (Christian et al, 2020) showed that levels of the reference or SSR-SINE-exon 2 isoforms were not different based on sex (Fig S4D). Thus, the loss of the protein derived from the SSR-SINE-exon 2 isoform leads to a sexually dimorphic cerebellar transcriptional profile that cannot to be simply explained by differences in its expression in male or female brains. More importantly, the fact that most cerebellar genes were up-regulated suggests that this isoform may normally be involved in inhibiting rather than activating gene expression in females. The degree of probe overlap between samples can be found in Fig S4E; the list of the 1,615 genes encompassed by these probes can be found in Table S2.

We submitted the differentially expressed genes (DEGs) to unsupervised hierarchical clustering to define patterns of expression, which revealed 4 clusters (Fig 3C, clusters I through IV). The largest differences were observed in female cerebella, involving genes within clusters I and II that were all up-regulated (257 and 677, respectively, Fig 3C). These genes enriched for processes specific to the brain, including receptors, transporters, and biosynthetic enzymes of the neurotransmitters glutamate, dopamine, serotonin, and cholinergic synapses (Table S2). RT-PCR confirmed the up-regulation of some of these targets (Fig 3D); the correspondence of

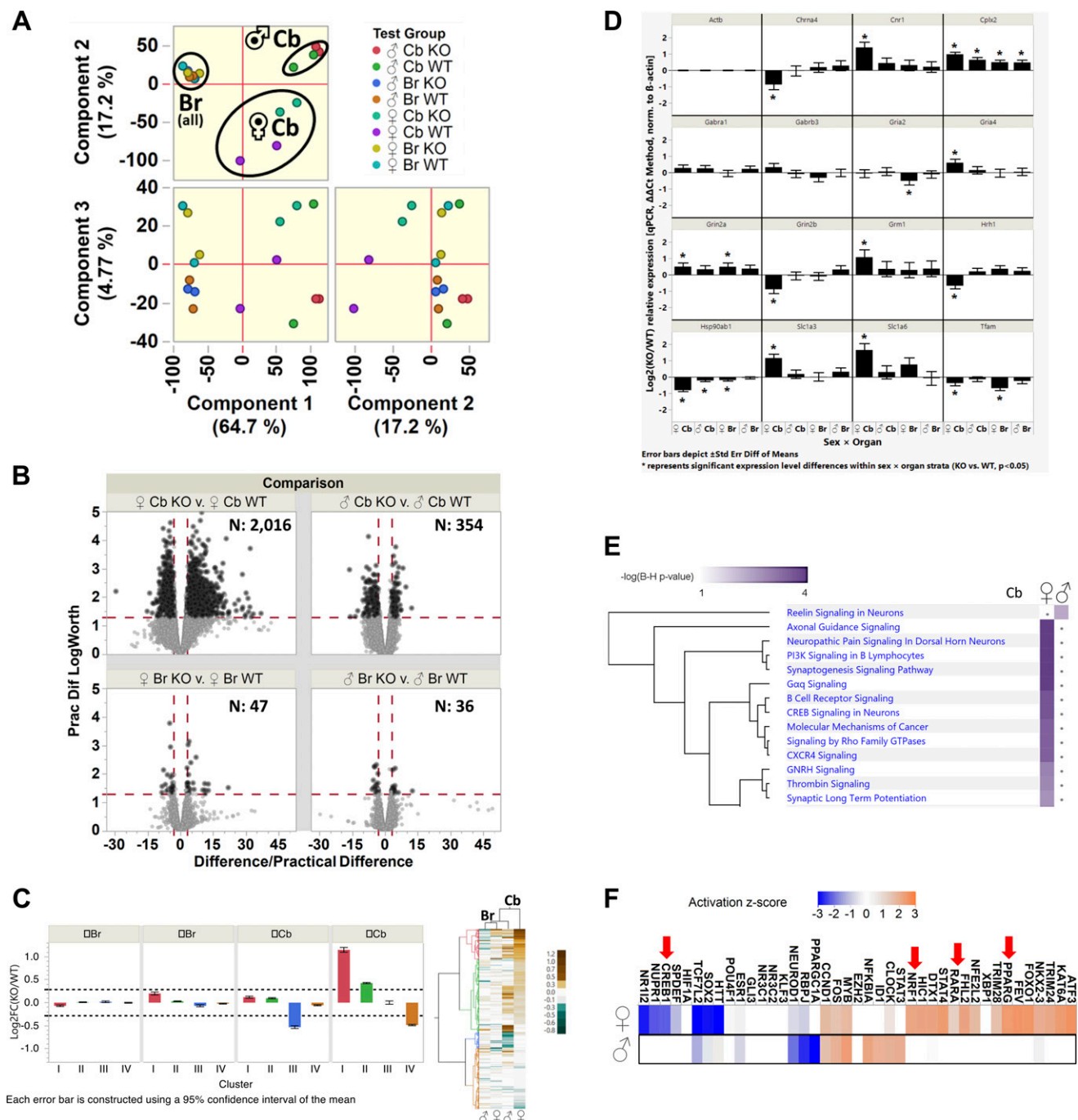

**Figure 3. Loss of the short interspersed nuclear element isoform leads to sexual dimorphic gene expression profiles.**
**(A)** Principal component analysis of 12,527 multivariate significant probe sets (RMA-normalized probe intensities, log₂FC ANOVA FDR $P < 0.05$, |log₂FC| > 0.286 for SNR > 1 versus grand mean; $PC1_{organ} + PC2_{sex} = 82\%$ total variance) was used to determine the main components driving the differences between gene expression profiles of wild-type and mutant littermates. **(B)** Volcano plots depicting 9,996 statistically differential probe sets between any genotype × organ × sex groups overall (multivariate significant, log₂FC post hoc pairwise $P < 0.05$; 5% practical difference |log₂FC| > 0.094); black highlights 2,363 differential probe sets combined that meet differential criteria in same-organ, same-sex comparisons between WT versus short interspersed nuclear element KO mice, the number of which is indicated inside each panel. **(C)** Gene expression data were segregated into patterns using unsupervised hierarchical clustering based on 1,980 single-gene probe sets encompassing 1,615 gene annotations, corresponding to the subset from same-sex 2,363 differentially expressed probes curated against RIKEN cDNA clones, multi-gene, or no-gene annotations. The heat map on the right (Cb, cerebellum; Br, rest of brain, that is, whole brain minus cerebellum) depicts the patterns giving rise to the distinct clusters: red cluster I (330 single-gene probe sets, 257 genes, log₂FC min: +0.12, max: +3.48, IQR: [+0.84, +1.31]), green cluster II (807 single-gene probe sets, 677 genes, log₂FC min: −0.32, max: +0.92, IQR: [+0.34, +0.55]), blue cluster III (172 single-gene probe sets, 156 genes, log₂FC min: −1.53, max: −0.10, IQR: [−0.63, −0.38]), and orange cluster IV (671 single-gene probe sets, 591 genes, log₂FC min: −2.79, max: +0.27, IQR: [−0.57, −0.37]); cluster-wise expression differences were deemed statistically robust based on |log₂FC| > 0.286 (SNR > 1 threshold). **(D)** RT-PCR was used to confirm the microarray identified glutamate-associated changes. **(E)** Ingenuity Pathway Analysis based on the curated list of 1,980 single-gene differential probe sets with statistically robust expression differences per group (|log₂FC| > 0.286; male cerebellum: 457 probe sets; female cerebellum:

microarray versus RT-PCR results is shown in Fig S4F. The second largest changes occurred in genes binned into clusters III in males (156) and IV in females (591), which were down-regulated (Fig 3C). Down-regulated genes included kinases, phospholipases, transporters, and channels in addition to immune-associated genes. No mitochondrial, antioxidant, or other genes, including complexin 1 (cplx1), which were previously identified as inhibited in the cerebellum of animals with the conditional exon 3 deletion allele in the brain (Lucas et al, 2014b), were identified by microarrays. Notably, expression of cplx2, a paralog of cplx1, was increased in our SINE KO mutants. Cplx2 is involved in negatively regulating the formation of synaptic vesicle clustering at the presynaptic membrane in post-mitotic neurons (Table S2). This increase, however, was not restricted to the cerebellum of females but was found to be significantly different in all SINE KO samples (Fig 3D). The lack of overlap in transcriptional targets likely reflects the differential effects associated with the SINE KO allele versus the complete loss of PGC1α in the brain with the deletion of exon 3 allele (Lucas et al, 2014b). These findings support the notion that the protein derived from the SINE isoform is not functionally equivalent in the brain to that encoded by the reference gene.

Next, we used Ingenuity Pathway Analysis (IPA) to identify biological processes enriched based on the cerebellar gene expression profiles. Given the large difference in the number of genes and their sex-specific up or down-regulation, we performed IPA separately in males and females. Several pathways relevant to brain physiology were enriched in females, whereas only reelin signalling was identified in males (Fig 3E). Interestingly, the first *reeler* KO mouse described showed severe cerebellar abnormalities (Caviness, 1976). We then used Kyoto Encyclopedia of Genes and Genomes (KEGG) to identify pathways enriched based on up-regulated versus down-regulated genes. We found more than 50 significantly represented KEGG pathways, many of them relevant to brain-specific processes including the first top 7 (Table S3), using up-regulated genes. The top category involved glutamatergic synapses, consistent with glutamate being one of the most abundant neurotransmitters in cerebellar cells (Zampini et al, 2016); up-regulation of glutamate-associated genes was confirmed by RT-PCR (Fig 3D). Increased glutamate signalling was recently reported in the neocortex and hippocampus of another conditional brain-specific PGC1α mutant involving exon 3 (McMeekin et al, 2020). In contrast, only three pathways were significantly enriched based on down-regulated genes, none of which were unique to brain physiology (Table S3).

The reference PGC1α isoform is known for interacting with a set of nuclear receptors (NRs) to regulate downstream targets in peripheral tissues, including the estrogen receptor α (ERα). Given the sexual dimorphism identified in the gene expression profiles, we asked whether estrogen through the ER could be involved in this response. Using HOMER, a motif analysis algorithm, and a window of ±1 Kb from the transcriptional start site of genes, we found that about 18% of DEGs had an estrogen responsive element (ERE);

notably, genes with ERE were significantly enriched among DEGs in the female versus the male cerebellum (Pearson $\chi^2$ $P$ < 0.0001; Table S2). KEGG analysis of these 290 genes demonstrated that they were enriched for four brain-specific pathways, although glutamatergic synapse was not among them (Table S4). To identify other potential proteins that could function as co-regulators of the gene expression with the protein expressed from the SSR-SINE-exon 2 isoform in an unbiased manner, we used IPA to predict the upstream drivers of the transcriptional program. IPA derives its prediction from established interactions between transcription factors (TFs) and target genes based on published experimental evidence. Using the female gene expression, we identified many TFs that were previously associated with stress response or inflammatory signalling (Fig 3F). It was noteworthy to find NRs known to interact with the reference PGC1α such peroxisome proliferator receptor gamma (PPARG), nuclear respiratory factor 1 (NRF1), the retinoic acid receptor α (RARα), and cyclic AMP responsive element binding protein 1 (CREB1) (Fig 3F). These NRs were unexpected because their classic downstream targets, such as those involved in the assembly of mitochondrial electron transport complexes or antioxidant enzymes, were not present within our dataset. Nevertheless, it is possible that these NRs regulate target genes in the brain that differ from those in peripheral tissues. For example, PPARG has been shown to modulate NF-κB immune-dependent gene expression in microglia (Bernardo & Minghetti, 2006), an effect that is not observed in the periphery.

To further explore the role of NRs in the female cerebellum, we used HOMER to identify the extent to which the identified DEGs harboured response elements (RE) that could be recognized by those NRs. We found that 20% of genes had a RE for NRF1, 13% for CREB1, and 2% for RARα (Table S2), but when using these genes, no pathway enriched with a significant adjusted $P$-value (Table S4). About 56% of DEGs had a PPARG binding site (Table S2), 457 of which were not only up-regulated but enriched for the same brain-specific pathways as the 884 up-regulated genes; glutamatergic synapse was the top category (compare Tables S3 and S4). Statistical analysis revealed that such a high enrichment for PPARG recognition sequence was not significantly different than the abundance of these sites in the other 13,509 transcribed but not differentially expressed cerebellar genes (Table S2). Nevertheless, genes with a PPARG binding site were significantly enriched among DEGs in the female versus the male cerebellum (Pearson $\chi^2$ $P$ < 0.0001; Table S2). Also, about 100 genes that we identified as having a PPARG RE using HOMER are predicted to be its downstream targets based on simulations on the PPAR database (Fang et al, 2016). Interestingly, PPARG has been shown to drive sexual dimorphic phenotypes in the periphery and in the brain (Duan et al, 2010; Park & Choi, 2017), including in models in which PPARG agonists were used (Benz et al, 2012). Finally, it is worth noting that CREB1 is required for the co-activation of PGC1α targets by participating in a complex that opens chromatin containing histone

---

1,738); dots indicate pathways without significant enrichment within gene sets per group. **(F)** Upstream regulators predicted based on the genes differentially expressed in the cerebellum of females (left) or males (right). Genes were analysed together or based on having or not a PGC1α recognition sequence within ±1 Kb of the annotate promoter. Z-scores range depicting activation (orange) or inhibition (blue) is shown.

acetyltransferase (HAT) activity, including p300 and the steroid receptor coactivator-1 (SRC-1; Puigsever et al, 1999). Loss of CREB1 in the brain has been shown to be reminiscent of HD (Mantamadiotis et al, 2002), a disease in which the homologous B1-B4 brain-specific PGC1α isoform has been implicated (Soyal et al, 2012).

## Discussion

Our understanding of the function of PGC1α in the brain is still emerging. Despite findings describing multiple and uniquely regulated transcriptional isoforms in the brain (Soyal et al, 2012, 2019, 2020), their function remain largely unknown. Full body and CNS-conditional *Pgc1α* KO mice were generated over a decade ago and found to have neurological phenotypes, including altered behaviour (Lin et al, 2004; Leone et al, 2005; Lucas et al, 2010, 2012, 2014b; Dougherty et al, 2014; McMeekin et al, 2018). Because those KO strains were generated by deletion of exon 3, which is common to all isoforms, it was impossible to dissect the potential contribution of the different *Pgc1α* brain transcripts to these phenotypes. In this study, by characterizing animals devoid solely of the SINE FT, we not only identified that loss of this isoform up-regulates genes but also that it drives a sex-dependent cerebellar transcriptional program. These findings suggest that the different brain *Pgc1α* isoforms are not functionally equivalent, with our data pointing that the protein expressed from the SSR-SINE-exon 2 isoform might not be involved in transcriptional co-activation. This may help explain conflicting reports about the benefits or detriment of modulating levels of the canonical PGC1α in the context of neurodegenerative disease (Ciron et al, 2012; Clark et al, 2012). Nevertheless, additional experiments, including the re-introduction of the reference isoform in the SINE KO mutants to define if it rescues the observed phenotypes would be required to probe functional equivalence.

Both similarities and differences were identified between the phenotypes previously described for the exon 3 deletion mutants (either the whole-body KO or CNS tissue-specific mutants) and those found with the SINE KOs we generated; differences were most prominent. For example, significant motor impairments were identified in both models, whereas gross neuroanatomical changes were unique to the exon 3 deletion mutants (Lin et al, 2004; Lucas et al, 2012). Despite this fact, the SINE mutants showed severe rotarod defects. This have led us to speculate that: (i) the protein derived from the SINE-containing isoform is associated with the rotarod defects, (ii) the loss of PGC1α in neurons is likely responsible for the motor deficits, (iii) the reference form of PGC1α, which is still expressed in the SINE KO mutants, does not compensate for the loss of the protein expressed by the SSR-SINE-exon 2 transcript, and (iv) that the gross anatomical brain lesions associated with the exon 3 deletion allele are not causing the motor defects (Lucas et al, 2012). However, we acknowledge that additional experiments are needed, including the generation of a highly specific antibody to the form of the protein being expressed from the SSR-SINE-exon 2 isoform, which will make possible to probe the expression of the various PGC1α protein isoforms in a cell type- or region-specific manner in the brain.

Another feature that was unique to the SINE KOs was the significantly altered gene expression program in the female cerebellum, which consisted primarily of up-regulation of genes, including those associated with neurotransmission. Sexual dimorphic effects had not been previously reported for the CNS-specific exon 3 deletion allele, but, to our knowledge, only males were studied in those experiments. The underlying cause for the sexual dimorphism in gene expression observed herein remains unclear but an obvious explanation could stem from the expression of sex-related hormones. In this context, our analysis identified that 20% of the DEGs are potential targets of ERα, which is activated upon estrogen binding. It is noteworthy that the distribution of ERα was reported to be different between brain regions and cell types in males and females (Simerly et al, 1997; Gillies & McArthur, 2010). Also, a recent study showed that sexual dimorphism in tissues, including in the brain, is a manifestation of different regulatory programs in males and females (Lopes-Ramos et al, 2020). Therefore, loss of one of the same interacting partners might have different outcomes depending on sex. Whether the protein expressed from the SSR-SINE-exon 2 isoform and ERα physically interact in the brain and the extent to which its loss might lead to differential gene expression in males versus females (and within different brain regions/cell types) requires further studies.

Interestingly, 62% of the genes containing an ERE in our dataset also bear a PPARG recognition sequence (Table S2), suggesting that some of those genes may be regulated by PPARG. This is consistent with previous reports demonstrating that PPARG can mediate the expression of estrogen target genes (Keller et al, 1995; Nunez et al, 1997). Recent data demonstrated that PPARG is highly expressed in neurons in the adult mouse brain (Warden et al, 2016), but little is still known about its downstream targets in this cell type. In fact, the effects of PPARG on brain physiology, other than neuroinflammation, are poorly understood and have been mostly inferred using agonists such as pioglitazone. Notably, this drug was shown to improve neurological deficits in different disorders, including rotarod performance in a mouse model of AD (Toba et al, 2016). Additional experiments are required to test the crosstalk between PPARG, the ERα or estrogen and the protein expressed from the SSR-SINE-exon 2 isoform of *PGC1α* in the brain.

The comprehensive up-regulation of gene expression like we identified here had not been reported with the exon 3 deletion mutants, which remove all brain isoforms. Thus, we concluded that the function of the SSR-SINE-exon 2 FT in gene expression is unlike that of the reference isoform of the gene. Nonetheless, other simpler explanations could justify the differences. For instance, only a few target genes were monitored in the brain of the exon 3 deletion mutant using RT-PCR (Lucas et al, 2010, 2012, 2014b), whereas we more broadly profiled this tissue using microarrays. Alternatively, the focus on down-regulated genes could have biased the analysis of the data from the exon 3 deletion mutant, which has largely been driven by the notion that PGC1α acts as a coactivator of gene transcription in the CNS (Lucas et al, 2010, 2012, 2014b; McMeekin et al, 2020). Interestingly, recent studies showed that the loss of PGC1α in the brain led to the up-regulation of genes,

including in the striatum and in the hippocampus, but only the down-regulated genes were studied (McMeekin et al, 2018). Our analysis of the striatum data using different statistical criteria showed that more than 1,000 genes were up-regulated, enriching for some of the same brain-relevant pathways as identified by us using the SINE-KO mutant animals. About 50% of them had a PPARG recognition element (Table S5).

Glutamatergic synapse was the top category enriched by the up-regulated genes in our study (Table S3). Although glutamate-associated genes have not been reported to be regulated by PGC-1α in the brain, a recent study found enhanced glutamatergic transmission in the neocortex and hippocampus of the exon 3 deletion mutant, indicating a role for PGC-1α in excitatory neurons (McMeekin et al, 2020). This would be consistent with our data in the cerebellum, in which granule cells are the most abundant glutamatergic neurons. It could also help explain the somewhat puzzling results that the conditional deletion of PGC1α in parvalbumin-expressing inhibitory interneurons does not produce motor deficits (Lucas et al, 2014a). Thus, it is tempting to speculate that the SSR-SINE-exon 2 isoform may normally act to co-repress genes in excitatory neurons, in this case glutamatergic, to support motor behaviour–an effect that would not be observed when the protein is absent in inhibitory cells. Experiments to determine which cells in the brain express the SSR-SINE-exon 2 isoform or if its expression is dependent on a specific type of neuronal function will help address this possibility.

Although the exact function of the SINE FT remains to be experimentally determined, it is worth noting that the protein expressed from this transcript has six amino acids, encoded by the SINE, that replace the 16 residues encoded by exon 1. Prediction of secondary structure reveals that this small amino acid change alters the protein structurally in that two α helices are replaced by a single larger α helix (Fig S5). Potentially, this may alter the ability of this isoform to dock with NRs in a way that prevents the recruitment of HATs, thus impairing transcription. In line with this possibility is our findings that CREB1-regulated gene expression is inhibited (Fig 3F); CREB1 is part of the HAT complex required for PGC1α to co-activate transcription (Puigserver et al, 1999). Alternatively, this altered N-terminus may recruit a histone deacetylase or "sequester" partners of (yet unidentified) TFs, which in turn inhibit their function. This would be like the effects of PGC1 α/PPARG in NF-kB signalling. Also, alterations in the amino-terminus of the protein may activate a transcriptional repressor for the genes that we find up-regulated. Clearly, additional work is required to understand how the protein expressed from the SSR-SINE-exon 2 isoform can ultimately impact gene expression regulation and how this may lead to aberrant biochemical functions in the brain.

Last, HIF1α was recently shown to activate transcription from the human B1 but not the reference promoter (Soyal et al, 2020). Although the role of HIF1α in the brain has previously been limited to hypoxic insult, a recent study has shown that crucial polarity-controlled events in neuronal determination and cerebellar germinal zone exit—including spindle orientation during neural stem cell division, axon-dendrite specification, and synaptogenesis (Singh & Solecki, 2015; Singh et al, 2016; Uzquiano et al, 2018)—are

sensitive to $O_2$ tension and thus could be regulated through HIF1α-dependent pathways (Kullmann et al, 2020). Whether the protein expressed from the SSR-SINE-exon 2 isoform participates in such events by interacting with HIF1α remains to be determined. Likewise, whether it plays a role in the physiological outcomes associated with brain ischemia, including those related to stroke and pre-term birth, are areas to be explored.

In summary, we established that the SSR-SINE-exon 2 FT isoform of *Pgc1α* is functional in vivo with roles in brain physiology that seems to differ from those of the reference isoform of the gene. The extent to which this isoform influences neurological disease, how it interplays with the other isoforms of PGC1α in different cell types in the brain, and whether its interaction with the ER or PPARG contributes to sexual dimorphic phenotypes that influence psychiatric disease constitute promising areas for future experimentation.

# Materials and Methods

### SINE-mutant animals

C57B6/J mice were purchased from The Jackson Laboratories. A single CAS9 target site (AATTGGAGCCCCATGGATGAAGG) was used to disrupt the ORF of the SINE-*Pgc1α* (SINE-Ppargc1a) variant. Complementary oligos were ordered from IDTDNA and cloned into a T7 sgRNA plasmid, and in vitro transcribed using Epicentre Ampli-Scribe T7 High Yield Transcription Kit. C57BL/6J one-cell embryos were microinjected with CAS9 SgRNA (10 ng/μl each) and 5′ capped/polyA tailed Cas9 RNA (100 ng/μl) derived from pCAG-T3-hCAS-pA, a gift from Wataru Fujii & Kunihiko Naito (Fujii et al, 2013). Microinjected embryos were surgically transferred to SWISS pseudo-pregnant females. At weaning, potential founders were screened by PCR amplicon sequencing (FWD: 5′-TGAGAATATCAGTCTCTGGGGGA-3′; Rev 5′-CAGCCCCTCCTCTGAAATACAAA-3′). Based on computationally predicted CAS9 off-target sites, the nearest genetically linked off-target site was nearly 27 Mb away and contained four mismatches to the CAS9 target sequence, and therefore was not screened in the founder mice. Founders of interest were bred to wild-type C57BL/6 mice and F1 offspring were re-screened to confirm germline transmission. The mutant mouse line was crossed to wild-type C57BL/6J mice for at least two generations to eliminate any unknown, unlinked mutations. Phenotyping was performed with founder line 4, which has a 4-bp deletion (TGAA) just 3′ of the SINE variant translational start site corresponding to chr5:51,912,715-51,912,718 (GRCm38/mm10 assembly). Mouse colony genotyping was performed by primer/probe assay by Transnetyx (FWD-Primer 5′-AGGTTTTTTGCGAAAATCAGTGAACTAAT-3′; REV-Primer 5′-GCAGTTTG-GAGCAATAGAGAAGAAC-3′; WT-PROBE 5′-AAAGTACCCTTCATCCATG-3′; Mutant-PROBE 5′-ACTTACAAAGTACCCTCCATG-3′). All animal protocols were approved by the Animal Care and Use Committee (ACUC) at the National Institute of Environmental Health Sciences (NIEHS) and experiments conducted in accordance with relevant guidelines and regulations. Female and male mice were included in all experiments, which were performed on age-matched WT and SINE homozygous littermates.

## PacBio sequencing and data analysis

We used RACE (Rapid Amplification of cDNA Ends) and whole brain RNA from the mouse to generate material for PacBio sequencing, which was performed at the National Institute Sequencing Core in Bethesda. After sequencing, quality control was performed with FastQC (Available online at https://www.bioinformatics.babraham.ac.uk/projects/fastqc/). Primers and the first 50 bases with non-uniform nucleotide composition were removed with Trimmomatic (Bolger et al, 2014). A Phred quality filter was also applied, keeping sequences with quality over 10 (Q ≥ 10). To identify *Pgc1α* transcript variants, we artificially generated template sequences containing all possible exon combinations (198) and aligned the PacBio reads to this reference sequence using Minimap2 tool (Li, 2018). Only alignments with map quality over 20 (MAPQ ≥ 20) that aligned through the junction points of the template sequences (Exon–Exon, SINE-Exon and SSR-SINE-Exon) were kept. The alignments were visually inspected using Integrative Genomics Viewer (Thorvaldsdottir et al, 2013).

## Antibody generation and specificity test

Antibodies were generated by Covance against the following amino acids of PGC1α: canonical isoform SQDSVWSDIEC, epitope MDEGYF within the SINE and NYGSSWETPSNQC within the SSR and those at position 513–526 at the C terminus of the protein. Serum was used for the experiments shown herein in a dilution of 1–10. Specificity of antibodies was judged with purified peptides and in NIH3T3 cells expressing HA-tagged recombinant *Pgc1α*, which was cloned using forward 5'-TTGACTGGCGTCATTCGGGA-3' and reverse 5'-TCAGGAA-GATCTGGGCAAAGAG-3' primers and expressed through the pInducer20 lentiviral vector (Addgene). Western blots were performed using actin as loading control; secondary antibodies were obtained from LI-COR and membranes were visualized using a LI-COR Odyssey imager.

## Histological analysis

Adult male and female SINE KO and WT littermate control mice were anesthetized with pentobarbital sodium and fixed via transcardial perfusion with 0.1 M PBS followed by 4% PFA in PBS. Brains were removed, rinsed in PBS and fixed overnight in 4% PFA at 4°C. Brains were rinsed in PBS before cryoprotection in 30% sucrose in PBS. Cryoprotected brains were embedded in tissue-freezing medium (Triangle Biomedical Sciences) and sectioned. For each genotype and sex n = 4 brains were sectioned at 25 $\mu$m in the sagittal or coronal plane and collected on Superfrost Plus microscope slides (Thermo Fisher Scientific). One set of slides was processed for 0.1% cresyl violet Nissl stain and a second with Luxol fast blue myelin stain. Following dehydration and clearing, slides were coverslipped with Permount (SP15-500; Thermo Fisher Scientific) mounting medium.

## Behavioural tests

For the behavioural battery, animals generated at NIEHS were shipped to the UNC Mouse Behavioural Phenotyping Laboratory where 15 male and 11 female WT controls and 13 male and 6 female

*Pgc1α* SINE isoform knockout (SINE KO) were tested. Mice were 14 wk old at the start of the behavioural testing. All animal care and procedures were conducted in strict compliance with the animal welfare policies set by the National Institutes of Health and by the University of North Carolina at Chapel Hill (UNC), and were approved by the UNC Institutional Animal Care and Use Committee.

## Elevated plus maze

This test was used to assess anxiety-like behaviour, based on a natural tendency of mice to actively explore a new environment, versus a fear of being in an open area. Mice were given one 5-min trial on the plus maze, which had two walled arms (the closed arms, 20 cm in height) and two open arms. The maze was elevated 50 cm from the floor, and the arms were 30 cm long. Mice were placed on the center section (8 × 8 cm), and allowed to freely explore the maze. Measures were taken of time on, and number of entries into, the open and closed arms.

## Open field test

Exploratory activity in a novel environment was assessed by a 1-h test in an open field chamber (41 × 41 × 30 cm) crossed by a grid of photobeams (VersaMax system, AccuScan Instruments). Counts were taken of the number of photobeams broken during the trial, and measures were taken of locomotion (total distance traveled), rearing movements, and time spent in the center region of the open field, an index of anxiety-like behaviour.

## Social approach in a three-chamber choice task

Mice were evaluated for the effects of *Pcg1α* SSR-SINE isoform deficiency on social preference. The procedure consisted of three 10-min phases: a habituation period, a test for sociability, and a test for social novelty preference. For the sociability assay, mice were given a choice between proximity to an unfamiliar, sex-matched C57BL/6J adult mouse ("stranger 1"), versus being alone. In the social novelty phase, mice were given a choice between the already-investigated stranger 1, versus a new unfamiliar mouse ("stranger 2"). The social testing apparatus was a rectangular, three-chambered box fabricated from clear Plexiglas. Dividing walls had doorways allowing access into each chamber. An automated image tracking system (Noldus Ethovision) provided measures of time in spent in each chamber and entries into each side of the social test box. At the start of the test, the mouse was placed in the middle chamber and allowed to explore for 10 min, with the doorways into the two side chambers open. After the habituation period, the test mouse was enclosed in the center compartment of the social test box, and stranger 1 was placed in one of the side chambers. The stranger mouse was enclosed in a small Plexiglas cage drilled with holes, which allowed nose contact. An identical empty Plexiglas cage was placed in the opposite side of the chamber. After placement of the stranger and the empty cage, the doors were re-opened, and the subject could explore the social test box for 10 min. At the end of the sociability

phase, stranger 2 was placed in the empty Plexiglas container, and the test mouse was given an additional 10 min to explore the social test box.

## Marble-burying assay for exploratory digging

Mice were given a 30-min test in a Plexiglas cage located in a sound-attenuating chamber with ceiling light and fan. The cage contained 5 cm of corncob bedding, with 20 black glass marbles (14 mm diameter) arranged in an equidistant 5 × 4 grid on top of the bedding. Measures were taken of the number of buried marbles (two thirds of the marble covered by the bedding).

## Acoustic startle test

This procedure was used to assess auditory function, reactivity to environmental stimuli, and sensorimotor gating. The test was based on the reflexive whole-body flinch, or startle response, that follows exposure to a sudden noise. Measures were taken of startle magnitude and prepulse inhibition, which occurs when a weak prestimulus leads to a reduced startle in response to a subsequent louder noise. Mice were placed into individual small Plexiglas cylinders within larger, sound-attenuating chambers. Each cylinder was seated upon a piezoelectric transducer, which allowed vibrations to be quantified and displayed on a computer (San Diego Instruments SR-Lab system). The chambers included a ceiling light, fan, and a loudspeaker for the acoustic stimuli. Background sound levels (70 dB) and calibration of the acoustic stimuli were confirmed with a digital sound level meter (San Diego Instruments). Each session began with a 5-min habituation period, followed by 42 trials. There were seven different types of trials: the no-stimulus (NoS) trials, trials with the acoustic startle stimulus (AS; 40 ms, 120 dB) alone, and trials in which a prepulse stimulus (20 ms; either 74, 78, 82, 86, or 90 dB) occurred 100 ms before the onset of the startle stimulus. Measures were taken of the startle amplitude for each trial across a 65-ms sampling window, and an overall analysis was performed for each subject's data for levels of prepulse inhibition at each prepulse sound level (calculated as 100 − [(response amplitude for prepulse stimulus and startle stimulus together/response amplitude for startle stimulus alone) × 100]).

## Buried food test for olfactory function

Several days before the olfactory test, an unfamiliar food (Froot Loops, Kellogg Co.) was placed overnight in the home cages of the mice. Observations of consumption were taken to ensure that the novel food was palatable. Sixteen to 20 h before the test, all food was removed from the home cage. On the day of the test, each mouse was placed in a large, clean tub cage (46 cm L × 23.5 cm W × 20 cm H), containing paper chip bedding (3 cm deep), and allowed to explore for 5 min. The mouse was removed from the cage, and one Froot Loop was buried in the cage bedding. Each subject was then returned to the cage and given 15 min to locate the buried food. Measures were taken of latency to find the food reward.

## Hot plate test for thermal sensitivity

Individual mice were placed in a tall plastic cylinder located on a hot plate, with a surface heated to 55°C (IITC Life Science, Inc.). Reactions to the heated surface, including hindpaw lick, vocalization, or jumping, led to immediate removal from the hot plate. Measures were taken of latency to respond. The maximum test length was 30 s, to avoid any type of paw damage.

## Morris water maze

The water maze was used to assess spatial and reversal learning, swimming ability, and vision. The water maze consisted of a large circular pool (diameter = 122 cm) partially filled with water (45 cm deep, 24–26°C), located in a room with numerous visual cues. The procedure involved two phases: a visible platform test and acquisition of spatial learning in the hidden platform task.

## Visible platform test

Each mouse was given four trials per day, across 2 d, to swim to an escape platform cued by a patterned cylinder extending above the surface of the water. For each trial, the mouse was placed in the pool at one of four possible locations (randomly ordered), and then given 60 s to find the visible platform. If the mouse found the platform, the trial ended, and the animal was allowed to remain 10 s on the platform before the next trial began. If the platform was not found, the mouse was placed on the platform for 10 s, and then given the next trial. Measures were taken of latency to find the platform and swimming speed via an automated tracking system (Ethovision 15, Noldus).

## Acquisition of spatial learning in the water maze via hidden platform task

After the visible platform task, mice were tested for their ability to find a submerged, hidden escape platform (diameter = 12 cm). Each animal was given four trials per day, with 1 min per trial, to swim to the hidden platform. Criterion for learning was an average group latency of 15 s or less to locate the platform, with a maximum of 9 d of training. After testing on Day 9, mice were given a 1-min probe trial in the pool with the platform removed. Selective quadrant search was evaluated by measuring the number of swim path crosses over the previous platform location, versus the corresponding locations in the other three quadrants.

## Fear conditioning

Animals were held in an anteroom separated from the testing room to ensure that the animals did not hear testing of other animals for at least 30 min before training/testing. Training took place in four identical sound-attenuating chambers (Context A; 28 × 21 × 21 cm; Med-Associates Inc.). The floor of each chamber consisted of a stainless-steel shock grid (1/2 inch apart) wired to a shock generator and scrambler (Med-Associates Inc.) to deliver foot shocks. Mice were evaluated for learning and memory in a conditioned fear test (Near-Infrared image tracking system, MED Associates). The

procedure had the following phases: training on Day 1, a test for context-dependent learning on Day 2, and a test for cue-dependent learning on Day 3. 2 wk after the first tests, mice were given second tests for retention of contextual and cue learning. Training: On Day 1, each mouse was placed in the test chamber, contained in a sound-attenuating box, and allowed to explore for 2 min. The mice were then exposed to a 30-s tone (80 dB) that co-terminated with a 2-s scrambled foot shock (0.4 mA). Mice received 2 additional shock-tone pairings, with 80 s between each pairing.

## Context- and cue-dependent learning

On Day 2, mice were placed back into the original conditioning chamber for a test of contextual learning. Levels of freezing (immobility) were determined across a 5-min session. On Day 3, mice were evaluated for associative learning to the auditory cue in another 5-min session. The conditioning chambers were modified using a Plexiglas insert to change the wall and floor surface, and a novel odour (dilute vanilla flavouring) was added to the sound-attenuating box. Mice were placed in the modified chamber and allowed to explore. After 2 min, the acoustic stimulus (80 dB tone) was presented for a 3-min period. Levels of freezing before and during the stimulus were obtained by the image tracking system. Second test rounds were conducted 2 wk after the first rounds.

## Accelerating rotarod, 5-min trials

At 16–19 wk in age, subjects were given two tests for motor coordination and learning on an accelerating rotarod (Ugo Basile, Stoelting Co.). The first test consisted of three trials, with 45 s between each trial. Two additional trials were given 48 h later. Retests were conducted when mice were 31–36 and 50–61 wk in age. Rpm (revolutions per minute) for each trial was set at an initial value of three, with a progressive increase to a maximum of 30 rpm across 5 min (the maximum trial length). Measures were taken for latency to fall from the top of the rotating barrel.

## Rapid-reversal rotarod test, 5-min trials

At 56–66 wk in age, subjects were given a two-trial retest for motor coordination, using a rapid-reversal procedure. Rpm (revolutions per minute) for each trial was fixed at 10 rpm. Reversal of the direction of barrel spin occurred approximately every 15 s across the trial (maximum 5 min).

## Accelerating and fixed speed test on rotarod, 1-min trial

At 4–6 mo of age in our initial cohort and at 32–37 wk of age in the behavioural test battery cohort, mice were evaluated in a rotarod procedure, modified from a previously described protocol (Lucas et al, 2012). Mice underwent two trials each at rotating speeds of 0–10 (accelerating), 16, 24, 28, and 32 fixed rpm. Each trial was a maximum of 60 s, with at least 5 min between each trial.

## Fixed speed test on the rotarod, 2-min trials

At 58–67 wk in age the behavioural test battery cohort, mice were evaluated in a final rotarod procedure, modified from a previously described protocol (Lucas et al, 2012). Mice underwent two trials each at rotating speeds of 24, 28, and 32 fixed rpm. Each trial was a maximum of 120 s, with at least 5 min between trials.

## Health status

Deficiency of the *Pgc1α* SINE isoform did not lead to overt changes in health or general motor ability. No subjects were lost from the behaviour study by the time of the final rotarod test, conducted when mice were age ~60–70 wk.

## Behavioural tests statistical analysis

For each procedure, measures were taken by an observer blind to mouse genotype. Behavioural data were analysed using one-way or repeated measures ANOVA, with separate analyses for males and females. Fisher's protected least-significant difference tests were used for comparing group means only when a significant F value was determined. Within-genotype repeated measures comparisons were used to determine side preference in the three-chamber choice test, and quadrant selectivity in the Morris water maze test. For all comparisons, significance was set at $P < 0.05$. Data presented in figures and tables are means (±SEM).

## Sample processing and RNA extraction

Mice were euthanized and their brains immediately removed. While fresh, extracted brains were manually cleaned of brain stem tissue remnants by gross dissection, and split into cerebellum and rest of brain (whole brain minus cerebellum). Each specimen was then stored in an individual 15-ml conical tube, placed on dry-ice to snap-freeze and archived at −80°C. To prevent cross-contamination and tissue degradation due to delayed processing, mice were dissected one at a time with a single-use scalpel blade each. RNA was extracted from the cerebellum or rest of the brain using TRIzol and 50–100 mg tissue per individual specimen; aqueous phase was retrieved to purify total RNA (ethanol precipitation, per manufacturer's guidelines). Aliquots with up to 1 µg total RNA were treated with DNAse I in solution (Invitrogen) before using for RT-PCR or microarrays.

## Microarrays, qRT-PCR, and data analysis

The Affymetrix Human Genome U133 Plus 2.0 GeneChip arrays were used to profile gene expression. Samples were prepared as per manufacturer's instructions. Arrays were scanned in an Affymetrix Scanner 3000 and data were obtained using the GeneChip Command Console and Expression Console Software (AGCC; Version 3.2 and Expression Console; Version 1.2) using the MAS5 algorithm to generate CHP-extension files. ANOVA was used to identify statistical differences between means of groups at $α < 0.05$ level among HG-U133 Plus 2.0 probe sets. Experiments followed a $2^3$ full-factorial design with N = 2 replication level, thus N = 8 per group (WT or

mutant) with two independent specimens each consisting of combinations across three variables (sex × genotype × organ) with two levels each (M versus F, WT versus Mut, Cb versus Rest-of-Br). All specimens were matched for litter set as much as possible (i.e., mouse origin's litters, parental breeding pair, or birth date). IPA was used to analyse differences between transcriptional profiles of SINE-mutant versus WT littermates based on differential expression readouts with SNR > 1 ($|\log_2 FC| > 0.286$) among the curated list of 1,980 single-gene differential probe sets (male cerebellum: 457; female cerebellum: 1,738; male rest-of-brain: 78; female rest-of-brain: 141). qRT-PCR was performed using the same microarray samples to assure reproducibility of the data as well as RNA obtained from an independent cohort of animals. A custom plate was purchased from Bio-Rad containing imprinted primers for the genes of interest depicted on Fig 3D; reactions were set using SYBR green following the manufacturer's protocol.

## Data Availability

Genomics data for this publication have been deposited in the National Center for Biotechnology Information (NCBI)'s Gene Expression Omnibus (Edgar et al, 2002; Barrett et al, 2011, 2013) and are accessible through Gene Expression Omnibus Series accession number GSE152224.

## Supplementary Information

## Acknowledgements

We thank the staff at the Core Facilities at NIEHS and NIH (Epigenetics and Genomics) and at UNC (Mouse Behavioral Phenotyping), Mr Greg J. Scott of the NIEHS Mouse Knockout Core for single cell embryo microinjection and surgical embryo transfer, and Ms Sydney Fry (NIEHS) for technical help with the brain images. We also thank Drs. Paul Wade, G. Jean Harry, and Kenneth Korach (NIEHS) for critical comments on the manuscript. This work was supported in part by the Intramural Research Program at the National Institute of Environmental Health Sciences of the National Institutes of Health (RPW) and by the Eunice Kennedy Shriver National Institute of Child Health and Human Development (U54 HD079124, SS Moy), a doctoral scholarship from the Chilean National Agency for Research and Development (ANID, No 21201090) to B Hernandez and by Fondecyt grant 11140869 to G Riadi (Chile).

### Author Contributions

OA Lozoya: data curation, formal analysis, investigation, and methodology.
F Xu: formal analysis, investigation, and methodology.
D Grenet: investigation and methodology.
T Wang: formal analysis.
KD: Stevanovic: investigation and methodology.
JD Cushman: investigation, methodology, and writing—review and editing.
TB Hagler: data curation, formal analysis, investigation, and methodology.
A Gruzdev: data curation, formal analysis, investigation, and methodology.
P Jensen: formal analysis, investigation, methodology, and writing—review and editing.
B Hernandez: formal analysis and methodology.
G Riadi: formal analysis and methodology.
SS Moy: formal analysis, investigation, and methodology.
JH Santos: conceptualization, data curation, formal analysis, supervision, writing—original draft, review, and editing.
RP Woychik: conceptualization, supervision, funding acquisition, and writing—review and editing.

### Conflict of Interest Statement

The authors declare that they have no conflict of interest.

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
