## [Reviewer comments · Life Science Alliance]

Life Science Alliance

A brain specific PGC1a fusion transcript affects gene expression and behavioral outcomes in mice

Oswaldo Lozoya, Fuhua Xu, Dagoberto Grenet, Tianyuan Wang, Korey Stevanovic, Jesse Cushman, Thomas Hagler, Artiom Gruzdev, Patricia Jensen, Bairon Hernandez, Gonzalo Riadi, Sheryl Moy, Janine Santos, and Richard Woychik

DOI: <https://doi.org/10.26508/lsa.202101122>

Corresponding author(s): Janine Santos, National Institute of Environmental Health Sciences and Richard Woychik, NIEHS

Review Timeline:

Submission Date:	2021-05-19
Editorial Decision:	2021-06-08
Revision Received:	2021-09-07
Editorial Decision:	2021-09-24
Revision Received:	2021-09-28
Accepted:	2021-09-29

Transaction Report:

June 8, 2021

Re: Life Science Alliance manuscript #LSA-2021-01122-T

Dr. Janine Santos
NIEHS
111 TW Alexander drive
Durham, NC 27709

Dear Dr. Santos,

Thank you for submitting your manuscript entitled "A brain specific PGC1a fusion transcript affects female gene expression and behavior outcomes in mice" to Life Science Alliance. The manuscript was assessed by expert reviewers, whose comments are appended to this letter. We invite you to submit a revised manuscript.

Thank you for this interesting contribution to Life Science Alliance. We are looking forward to receiving your revised manuscript.

Sincerely,

B. MANUSCRIPT ORGANIZATION AND FORMATTING:

Reviewer #1 (Comments to the Authors (Required)):

This study by Lozoya et al provides evidence for 2 novel PGC1a isoforms (SSR-exon2 and the SSR-SINE-exon2), validates their putative promoter region (SSR) specifically in the brain and shows expression of SSR-SINE-exon2 mostly in neurons. Evidence is presented for active translation of the isoforms but the predicted difference in the N-terminal 16AA between SSR-SINE-exon2 and the classic form failed to allow production of an antibody for specific detection of the protein products. Notably, the established role of PGC1a in transcriptional control is linked to its N terminus and so the function of the novel isoforms is an intriguing puzzle. To test if SSR-SINE-exon2 is functional in vivo this isoform was specifically targeted in mouse. This mouse shows brain specific decrease in total PGC1a protein and some deficits in motor coordination and motor learning. This is associated with gender-specific gene expression outcomes in the cerebellum. Overall, it is an important and broadly interesting manuscript that also carefully discusses the limitations of the study. I recommend a few amendments in the presentation:

- Fig2E (Rotarod test) legend seems to lack a statement on the gender.
- Fig2B age scale is confusing. A brake is needed between ticks corresponding 30 and 60 wks. Would the body weight difference warrant some discussion?

-some language errors like in Line 160-161 "No postnatal lethality as reported with the exon 3 deletion mutants (Lin et al., 2004) was not observed".

Reviewer #2 (Comments to the Authors (Required)):

The manuscript by Lozoya and colleagues investigates the role of PGC1a isoforms in mice, focusing on the brain, where alternative isoforms, including SINE PGC1a appear to be more highly expressed relative to the reference PGC1a. Using complementary approaches, they describe the regional expression and the cell type specificity of the isoforms. They then generate mice with a frameshift 4bp deletion intended to abrogate SINE isoform protein expression. They characterize the phenotypes of these mice and the transcriptional profiles. The findings suggest that SINE isoform has, at least in part, different regulatory effects on gene expression than reference PGC1a and that several of such differences are sex specific, possibly influenced by estrogen receptor and estrogen. Overall, the manuscript describes interesting observations and increases our understanding of the role of PGC1a in different cell types. There are some aspects of the study that deserve attention, to potentially strengthen the interpretation of the results.

1. The interpretation of the effects of the 4bp deletion (KO) on isoform expression in the brain needs some clarification. The antibody used in Fig 2 does not distinguish between isoforms, because it is against the C-terminal common sequence. The reference PGC1a should be present in brain lysates, because it is expressed in glia and partially in neurons. The KO has less total PGC1a brain signal, because it lacks the SINE isoform. So, in theory, there should be no SINE isoform in the KO brain. However, it is unclear how this interpretation can be experimentally confirmed, since the mRNA is not destabilized by the frameshift mutation and the antibody used is not specific for the SINE isoform. This caveat must be acknowledged.
2. The number of animal of each sex used for the motor performance and behavioral experiments is not clearly indicated in the figure or the legend (Fig. 2). On the other hand, n=4 is indicated for the western blots, but no quantification of average band intensity is provided. If the n=4 applies for the motor and behavioral tests, it would likely have insufficient power to provide a conclusive interpretation. So, likely more mice were used, but I can't find this number.
3. It is unclear what exactly figure 3C and 3D are indicating. Do they intend to demonstrate lack of changes in neuronal counts or lack of pathology? Perhaps, these histological data could be better addressed or omitted.
4. There seems to be significant differences between the effects of PGC1a exon 3 deletion (i.e., KO of all isoforms) and the SINE isoform KO investigated here. The results and discussion describe the similarities, but at the same time highlight some differences. Clearly, one of the main findings is that the isoforms are not functionally redundant. However, this message would be clearer if the differences and similarities were outlined in a more organized manner.
5. In the conditional and constitutive PGC1a KO models previously investigated it was shown that PGC1a was lost in neurons, as also cited here. Since neurons appear to express mostly SINE isoform, the outcomes should have been very similar to those described in the present model, but that was not always the case. Please, describe and discuss the differences and similarities with those model and possible reasons in a more systematic manner.

6. The potential interplay between estrogen receptors, sex hormones and SINE isoform of PGC1a in regulating gene expression in a sex, tissue and neuronal type specific manner is very interesting, although still hypothetical. The discussion mentions this intriguing possibility, but it would benefit from a step-by-step description of the proposed regulatory mechanism to better delineate the proposed hypothesis. Investigating the role of estrogen and using estrogen receptor KO mice may be necessary to better understand this interplay, but probably beyond the scopes of the study.

Reviewer #3 (Comments to the Authors (Required)):

1. SHORT SUMMARY

In the article "A brain specific *pgc1 α* fusion transcript affects female gene expression and behavior outcomes in mice," the authors present data indicating the existence of brain-enriched isoforms of the transcriptional coactivator peroxisome proliferator-activated receptor gamma coactivator 1 α (PGC-1 α) and demonstrate that mutations of an upstream region cause behavioral phenotypes and alterations in cerebellar gene expression in female mice. In general, this is an important report of novel brain-specific isoforms and of a novel mouse model, which together provide important information regarding PGC-1 α biology in the brain. However, there are some important revisions/additional data that are required to instill confidence in the authors' interpretations of the data and validity of the model. Specifically, the western blot data for PGC-1 α should be presented with 3-4 biological replicates per group per sex, so that the variability in this model is documented. Furthermore, considering the small sample sizes for the transcriptional studies, a couple top hits should be validated with quantitative rt-PCR, along with a couple of previously documented PGC-1 α -dependent neuron-enriched genes.

2. COMMENTS ON MAIN FINDINGS AND SUGGESTIONS FOR ADDITIONAL DATA

Considering the proposed roles for PGC-1 α deficiency in neurodegeneration, it is very important to understand its function in the brain. This study presents important information about novel isoforms enriched in neurons and demonstrates that disruption in the translation of the SSR-SINE-exon2 brain-specific isoform recapitulates motor impairment phenotypes observed with nervous system-specific deletion of PGC-1 α . Especially interesting is the observation that this model shows a complete reduction in PGC-1 α protein in the brain but not liver, indicating that the SSR-SINE-exon2 isoform could be the predominant form in the brain. Considering the novelty of these observations and the mouse model, it is critical for the authors to rigorously validate this model by demonstrating protein knockdown in multiple biological replicates (male and female) and to show any data for model validation in the main figures (instead of supplementary figure 2).

The authors chose the cerebellum and whole brain hemispheres for transcriptional analyses. While the cerebellum makes sense based on the enrichment of PGC-1 α in this region and the high proportion of GABAergic neurons in the cerebellum, it is not clear why the authors chose to transcriptionally profile an entire hemisphere. Due to the enrichment of PGC-1 α in sparse neuronal populations in some regions, it is very likely that the authors missed critical changes in gene expression that could only be observed with regional or cell-type-specific resolution. For example, the most PGC-1 α -responsive gene reported for the brain is parvalbumin, but deficiencies in parvalbumin in PGC-1 α -deficient mice are restricted to the forebrain and would not be noticed in homogenates including the thalamus. Regional dissections of the cortex and a standard rt-pcr assay for parvalbumin would enable the authors to compare their new model to the whole body and cell-type-specific knockout models.

The authors state that "in tissue lysates those raised against the FTs were unable to detect one only protein. Antibodies for the C-terminus recognized a protein of the correct molecular weight of an engineered HA-tagged PGC1 α recombinant protein that we expressed in NIH3T3 cells (Fig. S1B and C). From here on, this antibody was the one used to define the presence or absence of the FTs-derived proteins in the brain." So - this is confusing - if the C-terminus-specific antibody is used, wouldn't this recognize all full-length PGC-1 α isoforms? On a similar note - the authors state in the discussion that "these findings suggest that the different brain PGC-1 α isoforms are not functionally equivalent, with our data pointing that the protein expressed from the SSR-SINE-exon2 isoform might not be involved in transcriptional coactivation." I am not sure how this can be claimed, since the Western blots demonstrate a reduction in all C-terminal-reactive PGC-1 α in the brain (meaning that it should act as a complete brain KO without any remaining PGC-1 α).

The authors conclude from their transcriptional data that (Lines 272-274) "These may reflect the loss of all isoforms of Pgc1 in the brain in that study relative to the SINE FT (this study), supporting the notion that the proteins derived from these two brain isoforms are not functionally equivalent in this tissue." This is not consistent with the demonstration in Fig 2 that all C-terminal-containing PGC-1 α is gone in the SINE KO brain. In theory, shouldn't some transcriptional differences be replicated? This raises concerns about the small sample size for the transcriptional studies (n=2/group) and the lack of validation of any of the observed changes using rt-PCR.

The authors state in lines 269-272 that "It was noteworthy that no mitochondrial, antioxidant or other genes previously identified as differentially expressed in the cerebellum of animals with the conditional exon 3 deletion allele (Lucas et al., 2014b) were identified (Table S2)." What about neuron-enriched transcripts like Cplx1? The list of altered transcripts was not made available to reviewers.

3. OTHER COMMENTS

Some of the previous literature regarding the roles for PGC-1 α in the brain has been misrepresented by the authors. The authors incorrectly report that "only few genes associated with brain-specific functions were found" in PGC-1 α knockout mice; in fact, the main conclusion made in the Lucas and McMeekin series of papers is that key genes involved in synchronous neurotransmitter release and neuronal integrity are disrupted in whole body and cell-type-specific deletion models (reviewed in PMID: 33572179). In general, nuclear-encoded mitochondrial genes were reduced only modestly in brain from whole body knockout mice, with the largest reductions observed in genes enriched in parvalbumin-positive neuronal populations.

The authors also state that "the N-terminus of PGC-1 α is thought to dictate its transcriptional targets" (lines 103-105). This is not necessarily true. Data from Lucas et al. (2014, J. Neurosci.) demonstrates that mice lacking all PGC-1 α protein (Lin/Spiegelman line) have a reduction in mitochondrial as well as synaptic genes, while mice still retaining N-terminal portion (Kelly line) can maintain the mitochondrial gene expression but not synaptic gene expression. This implies that the C-terminal region of the protein is required for the maintenance of brain-specific PGC-1 α targets. This is consistent with what is known about the interactions of the C-terminus with RNA splicing factors and members of the mediator/TRAP complex. This is not to say, of course, that differences in N-terminal sequences could not confer special transcriptional properties upon brain-specific PGC-1 α isoforms, just there is not much data to support this assumption.

The observation of no early postnatal lethality is important, because this suggests that peripheral tissues are not affected. The cause of this in the Lin line is a reduction in fat

differentiation/maturation, causing hypothermia and death in pups (this can be noted in the text).

It is important that these mice show deficits in the rotarod assay, consistent with previous reports of motor impairment in nervous system-specific deletion of PGC-1 α .

Contrary to what is stated in Line 190, PGC-1 α expression is not reduced in postmortem tissue of patients with schizophrenia (McMeekin 2016), but PGC-1 α -dependent transcripts are reduced (Syt2, Cplx1, Nefh), suggesting a dissociation between PGC-1 α and its dependent genes (potentially due to a reduction in NRF1 expression).

The authors are correct that it is puzzling that mice lacking PGC-1 α in parvalbumin-positive populations do not show motor defects. One interpretation of this observation is that the studies used a cre line with late onset (postnatal day 14-30) of recombination. It is possible that full manifestation of the motor phenotype requires the deletion of PGC-1 α in parvalbumin-positive populations earlier in development.

While it would have been informative for the authors to use in situ hybridization to localize the novel transcript to specific brain cell populations, that could be considered as beyond the scope of the current study. However, the predicted cell-type-specific distribution of the novel PGC-1 α isoform is certainly an important topic which should be covered in the discussion.

Reviewer #1 (Comments to the Authors (Required)):

This study by Lozoya et al provides evidence for 2 novel PGC1a isoforms (SSR-exon2 and the SSR-SINE-exon2), validates their putative promoter region (SSR) specifically in the brain and shows expression of SSR-SINE-exon2 mostly in neurons. Evidence is presented for active translation of the isoforms but the predicted difference in the N-terminal 16AA between SSR-SINE-exon2 and the classic form failed to allow production of an antibody for specific detection of the protein products. Notably, the established role of PGC1a in transcriptional control is linked to its N terminus and so the function of the novel isoforms is an intriguing puzzle. To test if SSR-SINE-exon2 is functional in vivo this isoform was specifically targeted in mouse. This mouse shows brain specific decrease in total PGC1a protein and some deficits in motor coordination and motor learning. This is associated with gender-specific gene expression outcomes in the cerebellum. Overall, it is an important and broadly interesting manuscript that also carefully discusses the limitations of the study. I recommend a few amendments in the presentation:

-Fig2E (Rotarod test) legend seems to lack a statement on the gender.

Response: we have corrected the figure legend to indicate the number of males and females used for these studies.

-Fig2B age scale is confusing. A brake is needed between ticks corresponding 30 and 60 wks. Would the body weight difference warrant some discussion?

Response: we have included a brackets in the graph as suggested by the reviewer.

-some language errors like in Line 160-161 "No postnatal lethality as reported with the exon 3 deletion mutants (Lin et al., 2004) was not observed".

Response: we thank the reviewer for pointing this out and have carefully revised the text to fix all language errors.

Reviewer #2 (Comments to the Authors (Required)):

The manuscript by Lozoya and colleagues investigates the role of PGC1a isoforms in mice, focusing on the brain, where alternative isoforms, including SINE PGC1a appear to be more highly expressed relative to the reference PGC1a. Using complementary approaches, they describe the regional expression and the cell type specificity of the isoforms. They then generate mice with a frameshift 4bp deletion intended to abrogate SINE isoform protein expression. They characterize the phenotypes of these mice and the transcriptional profiles. The findings suggest that SINE isoform has, at least in part, different regulatory effects on gene expression than reference PGC1a and that several of such differences are sex specific, possibly influenced by estrogen receptor and estrogen. Overall, the manuscript describes interesting observations and increases our understanding of the role of PGC1a in different cell types. There are some aspects of the study that deserve attention, to potentially strengthen the interpretation of the results.

1. The interpretation of the effects of the 4bp deletion (KO) on isoform expression in the brain needs some clarification. The antibody used in Fig 2 does not distinguish between isoforms, because it is against the C-terminal common sequence. The reference PGC1a should be present in brain lysates, because it is expressed in glia and partially in neurons. The KO has less total PGC1a brain signal, because

it lacks the SINE isoform. So, in theory, there should be no SINE isoform in the KO brain. However, it is unclear how this interpretation can be experimentally confirmed, since the mRNA is not destabilized by the frameshift mutation and the antibody used is not specific for the SINE isoform. This caveat must be acknowledged.

Response: the reviewer is correct that in theory there should be no SINE isoform in the KO brain. Nevertheless, without antibodies specific for that isoform, we cannot determine if this is indeed the case. Instead, we took a subtractive approach using the c-terminal antibody as it identifies all isoforms of PGC1 α . Specifically, we surmised that the loss of the SINE-isoform would be identified as a reduction in the total levels of the protein in the brain. This is shown more clearly in **the revised Fig. 2**, in which now we show data from another 3 animals per group/sex, making the total number of animals analyzed n=8/sex/genotype. A graph showing the quantification of all the Westerns has also been included in the figure. However, we agree with the reviewer's comments that unless a SINE-specific antibody is available, this interpretation must be cautioned as the data could still reflect a combined decrease of all isoforms of PGC1 upon mutation of the SINE. While this is certainly feasible and could explain similarities between our model and the exon 3-deletion mutant, it cannot solely account for the differences identified in the phenotypes of the CNS exon 3 deletion mutant and our animals. For example, the brain lesions may result from inactivation of all isoforms of PGC1 α . As per the reviewer's request, we have acknowledged this caveat in the revised manuscript in the first paragraph of the Discussion (**see Page 17**).

2. The number of s of each sex used for the motor performance and behavioral experiments is not clearly indicated in the figure or the legend (Fig. 2). On the other hand, n=4 is indicated for the western blots, but no quantification of average band intensity is provided. If the n=4 applies for the motor and behavioral tests, it would likely have insufficient power to provide a conclusive interpretation. So, likely more mice were used, but I can't find this number.

Response: we apologize if the number of animals utilized for the behavioral tests was not clearly indicated in the figure legend; we had reported it in the Methods section. As per the request, we have revised the legend to include the information. For Fig. 2E, KO n=18, 12 males and 6 females, WT n=15, 8 males and 6 females. For all other behavioral test: KO n=19, 13 males and 6 females, and WT n=26, 15 males and 11 females. Likewise, we now provide the quantification for the Western blot data as a graph, which included the additional animals that have been analyzed as per request of reviewer 3.

3. It is unclear what exactly figure 3C and 3D are indicating. Do they intend to demonstrate lack of changes in neuronal counts or lack of pathology? Perhaps, these histological data could be better addressed or omitted.

Response: the reviewer is correct in that the data is provided to demonstrate the lack of changes in the anatomy of the brain regions of interest, which is important since this is a key feature of the exon 3 deletion mutant and thus a critical difference between the two models. Considering the request from another reviewer to move that data from to the supplement, we have chosen to show it as supplementary material.

4. There seems to be significant differences between the effects of PGC1 α exon 3 deletion (i.e., KO of all isoforms) and the SINE isoform KO investigated here. The results and discussion describe the similarities, but at the same time highlight some differences. Clearly, one of the main findings is that the isoforms are

not functionally redundant. However, this message would be clearer if the differences and similarities were outlined in a more organized manner.

Response: as suggested, we have revised the text and organized the discussion to show differences and similarities (see Page 16, 17). We hope the new organization is in line with the request of the reviewer.

5. In the conditional and constitutive PGC1 α KO models previously investigated it was shown that PGC1 α was lost in neurons, as also cited here. Since neurons appear to express mostly SINE isoform, the outcomes should have been very similar to those described the present model, but that was not always the case. Please, describe and discuss the differences and similarities with those model and possible reasons in a more systematic manner.

Response: we have revised the text as requested.

6. The potential interplay between estrogen receptors, sex hormones and SINE isoform of PGC1 α in regulating gene expression in a sex, tissue and neuronal type specific manner is very interesting, although still hypothetical. The discussion mentions this intriguing possibility, but it would benefit from a step-by-step description of the proposed regulatory mechanism to better delineate the proposed hypothesis. Investigating the role of estrogen and using estrogen receptor KO mice may be necessary to better understand this interplay, but probably beyond the scopes of the study.

Response: we agree with the reviewer that investigating the role of estrogen will be important, including by crossing our animals to the ER KO animals; we also agree this is beyond the scope of the manuscript. As requested, we have included in the discussion (Page 18) how we envision that estrogen or ER α may be influencing a portion of the gene expression program in our animals.

Reviewer #3 (Comments to the Authors (Required)):

1. SHORT SUMMARY

In the article "A brain specific pgc1 α fusion transcript affects female gene expression and behavior outcomes in mice," the authors present data indicating the existence of brain-enriched isoforms of the transcriptional coactivator peroxisome proliferator-activated receptor gamma coactivator 1 α (PGC-1 α) and demonstrate that mutations of an upstream region cause behavioral phenotypes and alterations in cerebellar gene expression in female mice. In general, this is an important report of novel brain-specific isoforms and of a novel mouse model, which together provide important information regarding PGC-1 α biology in the brain. However, there are some important revisions/additional data that are required to instill confidence in the authors' interpretations of the data and validity of the model. Specifically, the western blot data for PGC-1 α should be presented with 3-4 biological replicates per group per sex, so that the variability in this model is documented. Furthermore, considering the small sample sizes for the transcriptional studies, a couple top hits should be validated with quantitative rt-PCR, along with a couple of previously documented PGC-1 α -dependent neuron-enriched genes.

2. COMMENTS ON MAIN FINDINGS AND SUGGESTIONS FOR ADDITIONAL DATA
Considering the proposed roles for PGC-1 α deficiency in neurodegeneration, it is very important to understand its function in the brain. This study presents important information about novel isoforms enriched in neurons and demonstrates that disruption in the translation of the SSR-SINE-exon2 brain-specific isoform recapitulates motor impairment phenotypes observed with nervous system-specific

deletion of PGC-1 α . Especially interesting is the observation that this model shows a complete reduction in PGC-1 α protein in the brain but not liver, indicating that the SSR-SINE-exon2 isoform could be the predominant form in the brain. Considering the novelty of these observations and the mouse model, it is critical for the authors to rigorously validate this model by demonstrating protein knockdown in multiple biological replicates (male and female) and to show any data for model validation in the main figures (instead of supplementary figure 2).

Response: we thank the reviewer for the comments and apologize for not including quantification of the original data as the Western blots were representative. Based on the request, we have performed analysis in brains from 6 independent animals that were used for the behavioral tests. We now include in the main figure additional blots showing data from 3 males and 3 females/genotype; the graph shows quantification of all data collected to date encompassing n=8/sex/genotype.

The authors chose the cerebellum and whole brain hemispheres for transcriptional analyses. While the cerebellum makes sense based on the enrichment of PGC-1 α in this region and the high proportion of GABAergic neurons in the cerebellum, it is not clear why the authors chose to transcriptionally profile an entire hemisphere. Due to the enrichment of PGC-1 α in sparse neuronal populations in some regions, it is very likely that the authors missed critical changes in gene expression that could only be observed with regional or cell-type-specific resolution. For example, the most PGC-1 α -responsive gene reported for the brain is parvalbumin, but deficiencies in parvalbumin in PGC-1 α -deficient mice are restricted to the forebrain and would not be noticed in homogenates including the thalamus. Regional dissections of the cortex and a standard rt-pcr assay for parvalbumin would enable the authors to compare their new model to the whole body and cell-type-specific knockout models.

Response: We fully agree with the reviewer that by profiling the rest of the brain (brain minus the cerebellum), we were likely to miss critical changes associated with brain/cell type specificity. The reason why dissections were not performed initially was because our goal was 'simply' to determine whether the SINE-containing isoform was functional *in vivo*. When we started these analyses, we did not anticipate finding significant differences between the cerebellum and the rest of the brain, which were rather included as a control that otherwise would have been discarded.

As rightly pointed out by the reviewer, an obvious next step would be to perform the regional dissections, which will be useful for comparison purposes (relative to the other models) and help better understand the function of this isoform of PGC1a to overall brain physiology. Unfortunately, with the beginning of the pandemic we were forced to terminate our animal colonies. While we considered re-starting the colonies from the cryopreserved embryos, it became obvious that to perform the suggested experiments would require several months. Given the novelty of our data and the potential contribution to the field, we have thus decided to revise the text to clearly state the point raised by the reviewer. Specifically, the following sentence was included on **Page 11**: "The lack of broader changes in the Br likely reflects the analyses of several brain regions at once, which can dilute region-specific effects, rather than a cerebellum-unique phenotype. Future experiments dissecting and profiling the transcriptome of individual brain regions can address this issue".

We hope that the reviewer will agree that under current circumstances, the cost of obtaining these data outweighs the benefits of making our findings available for the broader scientific community through publication of this study.

The authors state that "in tissue lysates those raised against the FTs were unable to detect one only protein. Antibodies for the C-terminus recognized a protein of the correct molecular weight of an engineered HA-tagged PGC1 α recombinant protein that we expressed in NIH3T3 cells (Fig. S1B and C). From here on, this antibody was the one used to define the presence or absence of the FTs-derived proteins in the brain." So - this is confusing - if the C-terminus-specific antibody is used, wouldn't this recognize all full-length PGC-1 α isoforms? On a similar note - the authors state in the discussion that "these findings suggest that the different brain PGC-1 α isoforms are not functionally equivalent, with our data pointing that the protein expressed from the SSR-SINE-exon2 isoform might not be involved in transcriptional coactivation." I am not sure how this can be claimed, since the Western blots demonstrate a reduction in all C-terminal-reactive PGC-1 α in the brain (meaning that it should act as a complete brain KO without any remaining PGC-1 α).

Response: we apologize for the confusion and have clarified the sentence in the main text accordingly. Indeed, the C-terminal antibody identifies all isoforms of PGC1 α . As such, we took a subtractive approach in that the loss of the SINE-isoform would be identified as a reduction of the total levels of the protein in the brain but not in other tissues. This is what is shown in Fig. 2. Considering these data and the differential phenotypes between the complete brain KO and our SINE mutants, we feel that our data justifies the speculation that the two proteins (reference PGC1 α and the SINE-containing isoform) are not functionally the same. However, we are aware and have included in the revised discussion that additional experiments are required to address this fully (**Page 17**). We have no reason to think that this 4 bp deletion, which is located nearly 200 kb upstream from the reference promoter, would in any way affect the expression of the reference mRNA or translation of that mRNA to form the reference protein of PGC1 α in the brain. We have no evidence that the 4 bp deletion affects expression of the reference gene in non-brain tissues. Once antibodies specific for the protein expressed from the SSR-SINE-exon2 isoform are available, it will be possible to determine the relative levels, cell types and brain regions in which the reference PGC1 as well as the SINE-containing isoform are present and their potential contribution to the different phenotypes.

The authors conclude from their transcriptional data that (Lines 272-274) "These may reflect the loss of all isoforms of Pgc1 in the brain in that study relative to the SINE FT (this study), supporting the notion that the proteins derived from these two brain isoforms are not functionally equivalent in this tissue." This is not consistent with the demonstration in Fig 2 that all C-terminal-containing PGC-1 α is gone in the SINE KO brain. In theory, shouldn't some transcriptional differences be replicated? This raises concerns about the small sample size for the transcriptional studies (n=2/group) and the lack of validation of any of the observed changes using rt-PCR.

Response: As in the original submission, we have concluded that the total PGC1 α protein is reduced and not completely abolished in our KO animals. We have now increased the number of animals analyzed and included quantification of all the changes (n=8/sex/genotype) in the revised **Figure 2A**. Also, we have now generated RT-PCR data validating the changes as requested by the reviewer (**see new Figure 3D**).

The authors state in lines 269-272 that "It was noteworthy that no mitochondrial, antioxidant or other genes previously identified as differentially expressed in the cerebellum of animals with the conditional exon 3 deletion allele (Lucas et al., 2014b) were identified (Table S2)." What about neuron-enriched transcripts like Cplx1? The list of altered transcripts was not made available to reviewers.

Response: we apologize if the reviewer was not able to find the supplemental tables that, because of the large size, were provided as hyperlinks in the last page of the manuscript. From these links, tables were accessible and would have provided the data showing that none of the genes differentially expressed in the cerebellum in the previous paper, including *Clpx1*, were identified as changed in our SINE mutants when using microarrays. Interestingly, in our animals the *Clpx1* paralog, *Clpx2*, was rather increased. We have added that information to the paper (**Page 13**), and have provided the required data using RT-PCR (**new Figure 3D**).

3. OTHER COMMENTS

Some of the previous literature regarding the roles for PGC-1 α in the brain has been misrepresented by the authors. The authors incorrectly report that "only few genes associated with brain-specific functions were found" in PGC-1 α knockout mice; in fact, the main conclusion made in the Lucas and McMeekin series of papers is that key genes involved in synchronous neurotransmitter release and neuronal integrity are disrupted in whole body and cell-type-specific deletion models (reviewed in PMID: 33572179). In general, nuclear-encoded mitochondrial genes were reduced only modestly in brain from whole body knockout mice, with the largest reductions observed in genes enriched in parvalbumin-positive neuronal populations. The authors also state that "the N-terminus of PGC-1 α is thought to dictate its transcriptional targets" (lines 103-105). This is not necessarily true. Data from Lucas et al. (2014, J. Neurosci.) demonstrates that mice lacking all PGC-1 α protein (Lin/Spiegelman line) have a reduction in mitochondrial as well as synaptic genes, while mice still retaining N-terminal portion (Kelly line) can maintain the mitochondrial gene expression but not synaptic gene expression. This implies that the C-terminal region of the protein is required for the maintenance of brain-specific PGC-1 α targets. This is consistent with what is known about the interactions of the C-terminus with RNA splicing factors and members of the mediator/TRAP complex. This is not to say, of course, that differences in N-terminal sequences could not confer special transcriptional properties upon brain-specific PGC-1 α isoforms, just there is not much data to support this assumption. The observation of no early postnatal lethality is important, because this suggests that peripheral tissues are not affected. The cause of this in the Lin line is a reduction in fat differentiation/maturation, causing hypothermia and death in pups (this can be noted in the text). It is important that these mice show deficits in the rotorod assay, consistent with previous reports of motor impairment in nervous system-specific deletion of PGC-1 α .

Response: We thank the reviewer for highlighting these points, which we have included in the revised text as appropriate.

*Contrary to what is stated in Line 190, PGC-1 α expression is not reduced in postmortem tissue of patients with schizophrenia (McMeekin 2016), but PGC-1 α -dependent transcripts are reduced (*Syt2*, *Cplx1*, *Nefh*), suggesting a dissociation between PGC-1 α and its dependent genes (potentially due to a reduction in *NRF1* expression).*

Response: We thank the reviewer for kindly pointing this out as this was the original intent of that sentence. We have rectified the text to portray the information faithfully and apologize for the unintended misrepresentation of the cited paper.

The authors are correct that it is puzzling that mice lacking PGC-1 α in parvalbumin-positive populations do not show motor defects. One interpretation of this observation is that the studies used a cre line with late onset (postnatal day 14-30) of recombination. It is possible that full manifestation of the motor phenotype requires the deletion of PGC-1 α in parvalbumin-positive populations earlier in development.

Response: We agree with the reviewer that this is a possibility. Nevertheless, we would like to clarify that in the context of our work, the above puzzling data was used to highlight the possibility that the different isoforms of PGC1a may have distinct functions in different neuronal populations. Specifically, that the SSR-SINE-exon2 isoform may normally act to co-repress genes in excitatory neurons, in this case glutamatergic, to support motor behaviour. This would not be expected to be the phenotype upon deletion of PGC1a in parvalbumin-positive populations given their inhibitory nature.

While it would have been informative for the authors to use in situ hybridization to localize the novel transcript to specific brain cell populations, that could be considered as beyond the scope of the current study. However, the predicted cell-type-specific distribution of the novel PGC-1 α isoform is certainly an important topic which should be covered in the discussion.

Response: we thank for the suggestion. However, in situ hybridization cannot be used to localize the novel transcript to cell specific populations given that the unique portion of the SINE-containing transcript, which can differentiate this from other isoforms of PGC1a, is the SINE sequence itself. SINEs by being repetitive sequences on the genome are present in multiple copies. As such, in situ hybridization techniques while allowing hybridization to a SINE-sequence, cannot infer from which locus the sequence is being expressed. When single cell RNA-sequencing becomes an approach with higher sequencing depth, it can be utilized for the purpose suggested. Nevertheless, as of now, without antibodies to differentiate the distinct isoforms and with limited sequencing techniques available, this question remains technically impossible to address. We hope that soon these limitations will be overcome so that these important experiments can be performed. Text on **Page 18** of the manuscript states has been included to make this point.

September 24, 2021

RE: Life Science Alliance Manuscript #LSA-2021-01122-TR

Dr. Janine Santos
National Institute of Environmental Health Sciences
111 TW Alexander drive
Durham, NC 27709

Dear Dr. Santos,

Thank you for submitting your revised manuscript entitled "A brain specific PGC1a fusion transcript affects gene expression and behavioral outcomes in mice". We would be happy to publish your paper in Life Science Alliance pending final revisions necessary to meet our formatting guidelines.

- please add the Twitter handle of your host institute/organization as well as your own or/and one of the authors in our system
- please note that titles in the system and manuscript file must match
- please make sure the author order in your manuscript and our system match
- please consult our manuscript preparation guidelines <https://www.life-science-alliance.org/manuscript-prep> and make sure your manuscript sections are in the correct order
- please note that Tables must be in editable .doc or excel format
- please use the [10 author names, et al.] format in your references (i.e. limit the author names to the first 10)
- please upload one figure per page if possible (currently one figure is split up into a few pages)
- please add your main, supplementary figure, and table legends to the main manuscript text after the references section
- please upload your supplementary figures as single files also
- there is a legend missing for figure S2
- we encourage you to revise the figure legend for figure S1 to match the actual figure regarding the panels. Update the callouts in the text accordingly
- please revise call outs for Figure 1 in the manuscript text
- please add callouts for Figure S3A-F to your main manuscript text

Figure check:

- Please indicate molecular weight next to each blot
- missing scale bars for figure S2D, E, please indicate size in the legend

A. FINAL FILES:

B. MANUSCRIPT ORGANIZATION AND FORMATTING:

****Reviews, decision letters, and point-by-point responses associated with peer-review at Life Science Alliance will be published online, alongside the manuscript. If you do want to opt out of having the reviewer reports and your point-by-point responses displayed, please let us know**

immediately.**

Sincerely,

Reviewer #2 (Comments to the Authors (Required)):

In the revised manuscript, the authors have adequately addressed the minor points raised in my review.

Reviewer #3 (Comments to the Authors (Required)):

The authors have addressed all previous concerns in the revised document.

September 29, 2021

RE: Life Science Alliance Manuscript #LSA-2021-01122-TRR

Dr. Janine H. Santos
National Institute of Environmental Health Sciences
111 TW Alexander drive
Durham, NC 27709

Dear Dr. Santos,

Thank you for submitting your Research Article entitled "A brain specific PGC1a fusion transcript affects gene expression and behavioral outcomes in mice". It is a pleasure to let you know that your manuscript is now accepted for publication in Life Science Alliance. Congratulations on this interesting work.

DISTRIBUTION OF MATERIALS:

Again, congratulations on a very nice paper. I hope you found the review process to be constructive and are pleased with how the manuscript was handled editorially. We look forward to future exciting submissions from your lab.

Sincerely,
